# Differential Smoothing Mitigates Sharpening and Improves LLM Reasoning

**Jingchu Gai** [* 1]  **Guanning Zeng** [* 2]  **Huaqing Zhang** [* 2]  **Aditi Raghunathan** [1]

## Abstract

It is widely recognized that reinforcement learning (RL) fine-tuning of large language models often leads to *diversity collapse*, where outputs lack variety. Prior work has proposed a range of heuristics to counteract this effect, but these methods are ad hoc: they frequently trade off correctness for diversity, their effectiveness varies across tasks, and in some cases they even contradict one another. In this work, we place these observations on a rigorous foundation. We first provide a formal proof of why RL fine-tuning exhibits diversity collapse via a selection and reinforcement bias. Next, we make a key observation that any reward modification to address diversity collapse only needs to be applied on the correct trajectories. Building directly on this analysis, we introduce a principled method—*differential smoothing*—that provably improves both correctness and diversity, outperforming vanilla RL as well as widely used entropy-based heuristics. Our theory precisely characterizes when existing heuristics help and why they fail, while showing that differential smoothing is universally superior. Extensive experiments with models from 1B to 7B parameters, across domains including CountDown and real-world mathematical reasoning, demonstrate consistent gains. Differential smoothing improves both Pass@1 and Pass@k in most evaluated settings, with up to 11.4% improvements on AIME25 dataset.

## 1. Introduction

Reinforcement learning (RL) has become a powerful technique for fine-tuning Large Language Models (LLMs), enhancing capabilities ranging from complex reasoning (Guo et al., 2025; Yu et al., 2026; Shao et al., 2024) to human preference alignment (Ouyang et al., 2022; Bai et al., 2022). However, this process is often plagued by a significant side effect: a collapse in generation diversity (Song et al., 2025b; Dang et al., 2025; Chen et al., 2026b; Zhao et al., 2025; He et al., 2025a). This degradation is empirically observed in metrics like Pass@$K$; RL-tuned models often show diminishing improvements for larger values of $K$ and can even underperform the original base model (He et al., 2025a; Cobbe et al., 2021; Chow et al., 2025; Chen et al., 2026a).

However, mitigating this diversity collapse is non-trivial and presents several challenges. Simple heuristics such as early stopping or high-temperature decoding may boost diversity and achieve higher Pass@K, but they frequently hurt Pass@1 performance. This points to an inherent tension between increasing diversity and preserving accuracy. Second, most existing methods lack robustness across settings. A striking example is entropy control, where some works recommend maximizing entropy to improve both Pass@1 and Pass@K, while others report that minimizing entropy can yield the same outcome. Our experiments confirm that prior techniques designed to enhance diversity fail to succeed broadly and often succeed only on the tasks for which they were originally developed.

Motivated by these limitations, our work has two primary goals: (1) to provide clarity on the seemingly contradictory effects of previous methods, and (2) to develop a principled method that robustly improves both correctness (Pass@1) and diversity (Pass@K) across a range of benchmarks.

Our analysis from first principles shows that RL fine-tuning introduces two biases, **selection bias** and **reinforcement bias**, which jointly cause diversity collapse within the correct trajectories. Selection bias arises because correct trajectories with high probability under the pre-trained model are more likely to be reinforced (Theorem 6). Reinforcement bias then arises because these same trajectories receive disproportionately larger updates (Theorem 7).

Leveraging the insights from our theoretical analysis, we propose a *simple but novel twist* to vanilla RL. The core of our method is that while we need to change the reward function on correct trajectories to prevent the diversity collapse, the incorrect trajectories can continue to use the original reward or even a modification that encourages "sharpening".

[1]Carnegie Mellon University, Pittsburgh, PA, USA. jgai@andrew.cmu.edu, raditi@cmu.edu [2]Tsinghua University, Beijing, China. {zgn21,zhanghq22}@mails.tsinghua.edu.cn. Correspondence to: Jingchu Gai <jgai@andrew.cmu.edu>.

*Proceedings of the 43rd International Conference on Machine Learning*, Seoul, South Korea. PMLR 306, 2026. Copyright 2026 by the author(s).

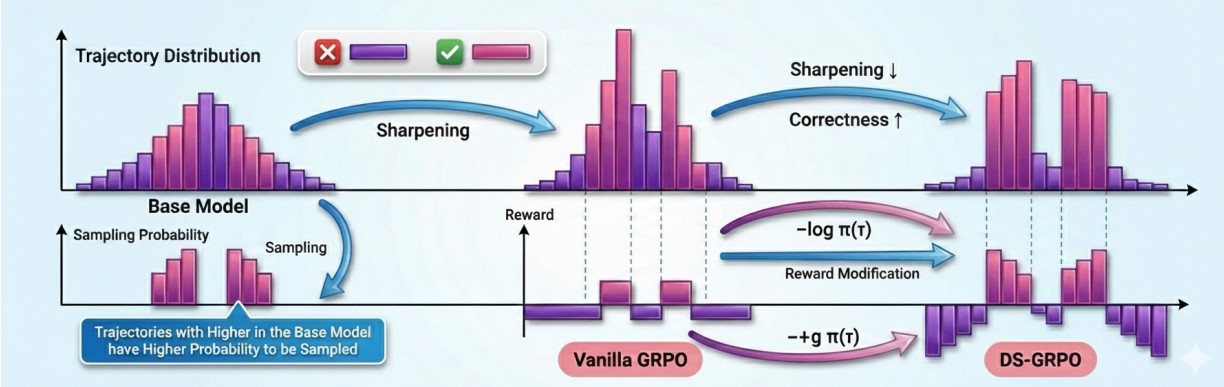

*Figure 1. Illustration on the effect of entropy*

This differentiated reward mechanism that applies distinct pressures to correct and incorrect trajectories can provably mitigate the tradeoff between correctness and diversity that plagues other heuristics.

We propose the **differential smoothing** approach. For correct trajectories, our reward mitigates the diversity collapse by subtracting a term proportional to their log-probability. On incorrect trajectories, our reward modification focuses on correctness, by adding the log-probability of the incorrect trace. We present our proposed DS-GRPO algorithm in Section 3.2.

We validate our differential smoothing approach both theoretically and empirically. Our theoretical analysis (Section 2) formally proves that the reward modification for correct trajectories directly optimizes for diversity, while the adjustment for incorrect ones enhances correctness without compromising diversity.

We evaluate DS-GRPO on five different models ranging from 1.5B-8B, and across a range of real-world settings, from simpler tasks such as Countdown to more challenging benchmarks in mathematical reasoning tasks. We consistently observe that DS-GRPO improves both Pass@1 and Pass@K relative to the vanilla baseline; on the Mistral-8B model, DS-GRPO outperforms the baseline by up to 11.4% (Pass@64 on AIME25). We further compare against prior heuristics, including entropy regularization and recent diversity-prompting techniques (He et al., 2025a; Chen et al., 2025b; Walder & Karkhanis, 2026). While these baselines yield improvements only in certain settings, DS-GRPO delivers robust gains across *most* datasets and models tested. Thus, DS-GRPO represents a principled approach that not only improves upon vanilla RL but also provides consistently stronger results than existing heuristics.

Finally, our analysis clarifies the contradictory effects of global entropy regularization in prior work. Increasing entropy across all trajectories improves diversity but reduces correctness, which can help on tasks with many valid solutions. Conversely, decreasing entropy improves correctness at the cost of diversity, which suits tasks with few valid

solutions. Our experiments confirm this principle, and we further show that *differential entropy control*, increasing entropy on correct trajectories while decreasing it on incorrect ones—achieves the best of both, paralleling the effect of differential smoothing.

> **Summary of Our Main Contribution**
>
> 1. We analyze diversity collapse from first principles in a formal setting.
>
> 2. Based on our diagnosis, we propose a novel differential smoothing algorithm that empirically improves both Pass@1 and Pass@K and outperforms previous methods in most of the real-world settings we tested.
>
> 3. We formally prove that our proposed differential smoothing approach improves diversity and correctness over the vanilla approach and the popular entropy maximization heuristic.
>
> 4. The analysis in this work also has broader implications of clarifying when and why existing heuristics work and guide the principled modifications of such heuristics.

## 2. A theoretical perspective on the diversity collapse in RL

In this section, we theoretically analyze the **diversity collapse** phenomenon that arises during the RL fine-tuning of LLMs where the fine-tuned model converges to a limited set of solutions for a given problem, neglecting other valid alternatives. (He et al., 2025a; Song et al., 2025a). This diversity collapse leads to a measurable degradation in metrics such as Pass@K as RL progresses, and potentially stifles the model's creative problem-solving capabilities.

Our analysis proceeds as follows: We first introduce our theoretical abstraction of the RL fine-tuning process (Sec 2.1). We then identify and analyze two primary factors that drive diversity collapse (Sec 2.2). Finally, based on these insights, we derive a principled reward function designed to miti-

gate this collapse and actively promote solution diversity (Sec 2.2 and Sec 2.4).

## 2.1. Setup

**Preliminaries.** We model language generation as a token-level Markov Decision Process (MDP). The environment is defined by a state space $\mathcal{S}$, a token vocabulary (action space) $\mathcal{A}$, and a maximum length $H$. An episode begins with an input prompt $\mathbf{x} \in \mathcal{X}$, which defines the initial state $s_1 = (\mathbf{x})$. At each step $h$, the state $s_h = (\mathbf{x}, a_1, \ldots, a_{h-1}) \in \mathcal{S}_h$ captures the prompt and previously generated tokens. The agent's policy $\pi_h(\cdot|s_h) : \mathcal{S}_h \rightarrow \Delta(\mathcal{A})$ provides a distribution over the next token. State transitions are deterministic: selecting action $a_h \in \mathcal{A}$ in state $s_h$ leads to $s_{h+1} = (\mathbf{x}, a_1, \ldots, a_h)$. A complete trajectory is $\tau = (\mathbf{x}, a_1, \ldots, a_H)$, which receives a terminal reward $r(\tau)$. Let $\mathcal{C}$ denote the set of all correct (successful) trajectories. We assume a binary reward function $r(\tau) = \mathbf{1}[\tau \in \mathcal{C}]$. We use an estimated reward function $\widehat{r}$, which is constructed as follows: Initially, $\widehat{r}(\tau) = 0$ for all $\tau$. A dataset of trajectories is sampled from base policy $\pi_{\text{base}}$. A verifier then identifies the subset of successful trajectories . The estimated rewards for these trajectories are subsequently set to 1.

**RL fine-tuning over a base policy.** A base model (or a pre-trained model) with corresponding policy $\pi_{\text{base}}$ is fine-tuned to optimize the following objective:

$$\pi_{\text{van}}(\tau) = \arg\max_{\pi} \mathbb{E}_{\tau \sim \pi} \widehat{r}(\tau) - \beta \cdot \mathbb{D}_{\text{KL}}(\pi || \pi_{\text{base}}), \quad (1)$$

where the KL-divergence term serves as a regularizer that prevents the updated policy from deviating too much from the base policy $\pi_{\text{base}}$ and $\beta$ is a hyperparameter that balances the trade-off between maximizing reward and preserving the knowledge of the base model. We denote this by $\pi_{\text{van}}$ to distinguish from proposed improvements in later sections.

## 2.2. Why does RL collapse diversity?

Ideally, RL fine-tuning should increase the likelihood of all correct trajectories. In practice, however, the process disproportionately reinforces correct trajectories that are already high-probability under the base LLM, a phenomenon we refer to as diversity collapse. As a starting point for mitigation, we first formally analyze *why* there is a diversity collapse in the theoretical setup discussed above.

---

### Analysis of Diversity Collapse

**Proposition 1** (**Selection bias**). *The probability that a correct trajectory's likelihood increases is monotonically related to its initial probability under the base model. Formally, for any two correct trajectories $\tau_1, \tau_2$ and $\beta > 0$, we have*

$$\pi_{base}(\tau_1) \geq \pi_{base}(\tau_2) \implies$$
$$\mathbb{P}(\pi_{van}(\tau_1) > \pi_{base}(\tau_1)) \geq \mathbb{P}(\pi_{van}(\tau_2) > \pi_{base}(\tau_2)).$$

**Proposition 2** (**Reinforcement bias**). *The magnitude of probability gain for a given corect trajectory is directly proportional to its probability under the base policy. Formally, if the reward update mechanism has access to the complete set of correct trajectories ($\widehat{r}(\tau) = 1$ for all correct trajectories), then for any correct trajectory $\tau$ and $\beta > 0$, we have*

$$\pi_{van}(\tau) - \pi_{base}(\tau) \propto \pi_{base}(\tau).$$

---

Proposition 1 reveals a *selection bias*: among correct trajectories, those with higher base probabilities are more likely to be reinforced. In addition, Proposition 2 shows that there is a *reinforcement bias*: these same high-probability correct trajectories further receive disproportionately larger boosts, further amplifying the model's existing preferences and sharpening the distribution. The proofs for Proposition 2 and Proposition 1 are provided in Appendix B.1. These results are derived by directly calculating the expression for the fine-tuned policy and analyzing its resulting probability distribution across trajectories. Crucially, the analysis of reinforcement and selection biases pertains specifically to correct trajectories. Conversely, incorrect trajectories exhibit an inverse dynamic: high-probability errors are subject to stronger penalization. Consequently, the probability distribution over incorrect trajectories is flattened rather than sharpened.

## 2.3. The correctness-diversity tradeoff

**A reward function to mitigate diversity collapse.** To counter diversity collapse, we modify the reward to discourage the reinforcement of high-probability trajectories. Since Lemma 5 implies that $\pi^{\star}(\tau) \propto \exp(\widehat{r}(\tau)/\beta)$, we define the modified reward $\tilde{r}(\tau) = \widehat{r}(\tau) - \gamma_{\text{ent}} \log \pi_{\text{base}}(\tau)$, where $\gamma_{\text{ent}} > 0$ controls the strength of regularization. The additional entropy bonus encourages the policy to assign more weight to trajectories that were less likely under the base policy countering the selection and reinforcement biases.

While such a change is natural, and has been previously proposed, we see that the gains of an entropy bonus are not universal. In Section 3.7 (Fig 5), we test out the entropy bonus objective and find that entropy bonus improves performance on Countdown but hurts performance (and underperforms standard GRPO) on MATH500.

Conversely, other works advocate for an *entropy penalty* i.e. adding rather than subtracting $\gamma_{\text{ent}} \log \pi_{\text{base}}(\tau)$ rather than

an entropy bonus to improve performance (Gao et al., 2025; Agarwal et al., 2026). We see that entropy penalty in fact improves performance on MATH500, while underperforming on Countdown.

**This apparent contradictory benefits of both entropy bonus and entropy penalty suggests the optimal strategy is task-dependent.** Intuitively, entropy bonus improves diversity but at the cost of correctness. Such a tradeoff has also been reported with other methods such as increasing temperature (Dang et al., 2025). On the other hand, entropy penalty does the reverse and improves correctness but at the cost of diversity. For tasks with high inherent diversity in correct solutions, the gain in diversity offers a significant benefit and an entropy bonus overall works out to be beneficial; for tasks with low solution diversity, the increase in diversity does not offer a benefit that outweighs a potential decrease in correctness, and an entropy penalty ends up being beneficial.

We can observe this more precisely by defining a *Solution Multiplicity* metric, defined as the average number of unique correct solutions per problem:

$$\text{Solution Multiplicity}(\mathcal{X}) = \frac{1}{|\mathcal{X}|} \sum_{x \in \mathcal{X}} A(x), \quad (2)$$

where $A(x)$ is the number of correct solutions for problem $x$.

We experimentally test the performance of entropy bonus/penalty on a few other datasets and find that entropy bonus is more effective when Solution Multiplicity is high, whereas an entropy penalty performs better when it is low (Section 3.7).

Overall, entropy bonus addresses diversity collapse but at the cost of correctness. Furthermore, both entropy maximization and minimization have been proposed as valid modifications, but neither is universally optimal across all tasks.

## 2.4. A differential approach to mitigating the tradeoff

The natural mitigation of diversity collapse—entropy bonus—is not the perfect solution because it decreases correctness.

*Is this tradeoff inherent, or is there a way to simultaneously improve both the correctness and diversity?*

We make a key observation: **we only care about diversity over correct (high reward) trajectories**. Also, reinforcement and selection biases pertains specifically to correct trajectories. Hence, one must redefine the desired diversity metric to focus exclusively on correct trajectories.

We formalize our definition in the simple theoretical framework described previously in Section 2.1. We then describe

a new differential reward modification that assigns different entropy terms to the positive and negative trajectories. Finally, we theoretically show that our proposed differential loss mitigates the tradeoff between diversity (over high-reward trajectories) and correctness.

**Definition 3** (Correctness and Correct-Solution Diversity). *For any policy $\pi$, we define its correctness as $C(\pi) = \sum_{\tau \in \mathcal{C}} \pi(\tau)$. We use the normalized variance on correct trajectories to measure diversity over correct solutions. Namely, we define $\sigma(\pi) = [\sum_{\tau \in \mathcal{C}} \pi(\tau)^2 |\mathcal{C}| - C(\pi)^2 / C(\pi)^2$.*

A small normalized variance indicates that the probability distribution is more uniform (i.e., less "sharp"). In such a distribution, the probability mass is not concentrated on a few high-probability trajectories but is spread more evenly across a wider set of solutions. This directly translates to higher diversity, as a greater variety of trajectories are likely to be sampled. Therefore, a policy with lower normalized variance exhibits higher diversity. Based on this insight, we consider the following differential reward modification:

$$r_{DS}(\tau) = \begin{cases} \widehat{r}(\tau) - \gamma_{\text{DS}} \cdot \log(\pi_{\text{base}}(\tau)) & \text{if } \widehat{r}(\tau) > 0 \\ \widehat{r}(\tau) & \text{if } \widehat{r}(\tau) \leq 0. \end{cases}$$
$$(3)$$

We then proceed to compare $\pi_{\text{DS}}$ that maximizes the reward above, and $\pi_{\text{ent}}$ that maximizes the standard entropy bonus modification to improve diversity.

$$\pi_{\text{ent}} = \arg\max_\pi \mathbb{E}_{\tau \sim \pi}[\tilde{r}(\tau)] - \beta_{\text{ent}} \cdot \mathbb{D}_{\text{KL}}(\pi || \pi_{\text{base}}),$$
$$\pi_{\text{DS}} = \arg\max_\pi \mathbb{E}_{\tau \sim \pi}[r_{\text{DS}}(\tau)] - \beta_{\text{DS}} \cdot \mathbb{D}_{\text{KL}}(\pi || \pi_{\text{base}}).$$

Here, $\beta_{\text{ent}}$ and $\beta_{\text{DS}}$ denote the KL coefficients for the two RL algorithms we are comparing, which use different reward modifications. With the definitions above, we aim to show that our differential reward modification performs better in both correctness and diversity. To this end, we present the following theoretical result:

---

**Theoretical Benefit of Differential Reward**

**Theorem 4.** *Assume the model have correct estimation for the reward of all trajectories. For any parameters $\gamma_{ent} \geq 0$ and $\beta_{ent} > 0$ used in Eq. 1 (for $\pi_{ent}$) that satisfy a proximity constraint $K_\rho(\pi_{ent}, \pi_{base}) \leq \kappa$, there exist parameters $\gamma_{DS} \geq 0$ and $\beta_{DS} > 0$ for $\pi_{DS}$ such that it also satisfies $K_\rho(\pi_{DS}, \pi_{base}) \leq \kappa$, and the following inequalities hold:*

$$C(\pi_{DS}) \geq C(\pi_{ent}) \quad and \quad \sigma(\pi_{DS}) \leq \sigma(\pi_{ent}).$$

*This result holds for $K_\rho(\pi, \pi_{base}) \in \{\mathbb{D}_{KL}(\pi || \pi_{base}), \mathbb{D}_{KL}(\pi_{base} || \pi), \mathbb{D}_{\chi^2}(\pi || \pi_{base}), \mathbb{D}_{\chi^2}(\pi_{base} || \pi)\}$.*

---

The KL-divergence constraint ($K_\rho(\cdot, \pi_{\text{base}}) \leq \kappa$) is a practical necessity and a standard assumption in prior work (Setlur et al., 2025). It prevents the fine-tuned policy from deviating excessively from the base model, thereby retaining pre-trained knowledge and avoiding catastrophic forgetting. Thus, Theorem 4 demonstrates that differential reward modification (for $\pi_{\text{DS}}$) achieves superior correctness ($C$) and diversity ($\sigma$) while adhering to the same proximity constraints as the global penalty method (for $\pi_{\text{ent}}$).

## 2.5. Differential Smoothing with Entropy Bonus and Entropy Penalty

We proved that increasing entropy on correct trajectories alone can mitigate the tradeoff with correctness. Additionally, our analysis indicates that reducing the entropy (diversity) of incorrect trajectories enhances model correctness without detriment to the diversity of correct generations. With this perspective, let's revisit entropy penalty which we saw improves performance on some tasks with low solution diversity such as MATH500 but underperformed on tasks with high solution diversity such Countdown. Interestingly, theorem 4 interestingly holds even if we apply an entropy penalty exclusively on negative trajectories. In other words, an entropy penalty on negative trajectories does not impact the policy's diversity over correct responses and hence such a modification could prevent

Motivated by this finding, as well as the empirical observation that entropy penalties can improve correctness (Gao et al., 2025; Agarwal et al., 2026), we propose our final method. We term this approach Differential Smoothing (DS), which applies an entropy bonus exclusively to positive samples and an entropy penalty exclusively to negative samples:

$$r_{\text{DS}}(\tau) = \begin{cases} \widehat{r}(\tau) - \gamma_p \cdot \log(\pi_{\text{base}}(\tau)) & \text{if } \widehat{r}(\tau) > 0 \\ \widehat{r}(\tau) + \gamma_n \cdot \log(\pi_{\text{base}}(\tau)) & \text{if } \widehat{r}(\tau) \leq 0, \end{cases} \quad (4)$$

where $\gamma_p, \gamma_n \geq 0$ are hyperparameters. This formulation generalizes the reward modification strategy in Equation (3). In the following sections, we first detail the implementation of differential smoothing in GRPO. We will then demonstrate empirically that differential smoothing achieves a significant advantage: it simultaneously increases both correctness (Pass@1) and diversity (Pass@K) across different tasks, outperforming other methods that aim to improve diversity.

# 3. Empirical evaluation of differential smoothing GRPO

In this section, we empirically evaluate the effectiveness of our proposed reward modification (Eq.4) in LLM reinforcement finetuning.

We first recap the popular GRPO algorithm for LLM fine-

tuning with RL and present our adaptation, DS-GRPO that incorporates the differential reward described in Section 2.5. We then compare DS-GRPO with GRPO on Countdown and mathematical reasoning. We further compare our proposed method with other methods aimed at improving diversity, including entropy-based approaches and other recently proposed techniques. Our key finding is that DS-GRPO consistently outperforms both GRPO and prior approaches on most tasks. In contrast, previous methods only outperform baselines on a subset of tasks because they fail to resolve the inherent tradeoff between correctness and diversity. We further analyze entropy-based approaches (both entropy bonus and entropy penalty) and show why their gains are uneven and fundamentally limited. In contrast, our differential reward formulation offers a principled way to enhance diversity over correct trajectories, leading to consistent improvements across the board.

## 3.1. Preliminaries: Group Relative Policy Optimization

We adopt *Group Relative Policy Optimization* (GRPO) (Shao et al., 2024) as our training backbone. For each input $x$ sampled from the training set, the policy decodes a group of $G$ completions $\{y_i\}_{i=1}^{G} \sim \pi_{\theta_{\text{old}}}(\cdot \mid x)$, where $\pi_{\theta_{\text{old}}}$ denotes the behavior policy used to collect the batch (the policy parameters at the previous update). Let $r_i = r(y_i)$ be the scalar reward of completion $y_i$, and denote by $\mu(\{r_j\}_{j=1}^{G})$ and $\sigma(\{r_j\}_{j=1}^{G})$ the mean and standard deviation of $\{r_j\}_{j=1}^{G}$, respectively. GRPO replaces the learned critic with a *group baseline* and uses the group-standardized advantage $A_i = [r_i - \mu(\{r_j\}_{j=1}^{G})]/\sigma(\{r_j\}_{j=1}^{G})$. GRPO adopts a clipped objective with a forward KL regularizer to the fixed *base policy* $\pi^b$: $\mathcal{J}_{\text{GRPO}}(\theta) = \mathbb{E}_x \mathbb{E}_{\{y_i\}_{i=1}^{G} \sim \pi_{\theta_{\text{old}}}(\cdot \mid q)} \big[ \frac{1}{G} \sum_{i=1}^{G} \frac{1}{|y_i|} \sum_{t=1}^{|y_i|} \min \{ \rho_{i,t}(\theta) A_i, \text{clip}(\rho_{i,t}(\theta), 1 - \epsilon, 1 + \epsilon) A_i \} - \beta_{\text{KL}} \mathbb{D}_{\text{KL}}(\pi_\theta \| \pi^b) \big]$ where $\rho_{i,t}(\theta) = \frac{\pi_\theta(y_{i,t} \mid x, y_{i,<t})}{\pi_{\theta_{\text{old}}}(y_{i,t} \mid x, y_{i,<t})}$ is the importance ratio.

## 3.2. Reward Modification for Mitigating Diversity Collapse

To operationalize the theoretical principles from Section 2.4 within the GRPO framework, we propose *Differential Smoothing GRPO* (DS-GRPO), a novel algorithm that reshapes the advantage function $A_i$ (Eq. 3.2). This reshaping directly implements our theoretical reward modification for a detailed discussion). Specifically, for successful completions (where $r_i = 1$), we subtract the term $\gamma_p \log \pi_{\theta_{\text{old}}}(y_i \mid x)$ from the advantage; conversely, for unsuccessful completions, we add the term $\gamma_n \log \pi_{\theta_{\text{old}}}(y_i \mid x)$.

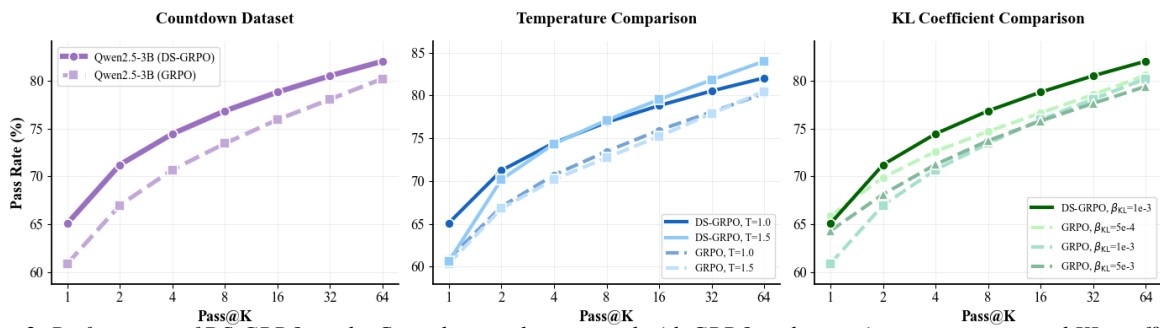

Figure 2. Performance of DS-GRPO on the Countdown task, compared with GRPO under varying temperatures and KL coefficients.

**Estimation of Entropy Discrepancy**

$$A_i^{\mathrm{DS}} = A_i \; + \; \begin{cases} - \gamma_p \, \log \pi_{\theta_{\mathrm{old}}}(y_i \mid x), & \text{if } r_i = 1, \\ + \gamma_n \, \log \pi_{\theta_{\mathrm{old}}}(y_i \mid x), & \text{otherwise}, \end{cases}$$

We plug the modified advantages $A_i^{\mathrm{DS}}$ into the GRPO objective:

$$\mathcal{J}_{\mathrm{GRPO}}(\theta) = \mathbb{E}_x \mathbb{E}_{\{y_i\}_{i=1}^G \sim \pi_{\theta_{\mathrm{old}}}(\cdot|q)} \Big[ \frac{1}{G} \sum_{i=1}^G \frac{1}{|y_i|} \sum_{t=1}^{|y_i|} \min\{ \rho_{i,t}(\theta) \, A_i^{\mathrm{DS}}, \, \mathrm{clip}(\rho_{i,t}(\theta), \, 1-\epsilon, \, 1+\epsilon) \, A_i^{\mathrm{DS}} \} \; - \; \beta_{\mathrm{KL}} \mathbb{D}_{\mathrm{KL}}(\pi_\theta \, \| \, \pi^b) \Big].$$

In our theoretical framework, we employ $\pi_{\mathrm{base}}$ as the bonus and penalty terms within the modified reward function. Notably, our analysis remains valid even if $\pi_{\mathrm{base}}$ is substituted with $\pi$. However, for practical implementation, we empirically select $\pi_{\theta_{\mathrm{old}}}$ due to its superior stability. A detailed justification linking the theory to this practical choice is provided in Appendix B.3. We evaluate DS-GRPO on the Countdown and MATH reasoning benchmarks across a range of models. As demonstrated in the subsequent sections, our method consistently improves both correctness (Pass@1) and diversity (Pass@K), outperforming existing diversity-promoting approaches on most evaluation metrics.

### 3.3. Experimental Setup

We detail our experimental setup below, covering task specifications, model architectures, and hyperparameter configurations.

**Countdown.** We evaluate the performance of DS-GRPO on the arithmetic reasoning task *Countdown*, where 3 to 4 integers in $[1, 99]$ are provided and the goal is to produce an equation that equals a target number using all given integers exactly once. Following Pan et al. (2025), we use the Qwen2.5-3B-Instruct(Hui et al., 2024) as the backbone model. For DS-GRPO, we set $\gamma_p = 0.03, \gamma_n = 0.01$. More experiment details are given in Appendix C.1.

**Math Reasoning.** We further explore DS-GRPO on the task of math reasoning. We choose four models with varied

sizes and response lengths, Qwen2.5-Math-1.5B (Hui et al., 2024), Qwen3-1.7B (Yang et al., 2025), Qwen2.5-Math-7B (Hui et al., 2024) and Ministral-8B-Instruct (Jiang et al., 2023). The three Qwen models are fine-tuned on 14,053 English questions filtered from the DAPO17k (Yu et al., 2026) dataset (DAPO14k); Ministral-8B-Instruct is additionally augmented with 12,000 questions from the MATH12k dataset (Hendrycks et al., 2021). We train the Qwen models with batch size 32 and Ministral with batch size 128 until the training reward saturates, then evaluate on five commonly used benchmarks of math reasoning: MATH500 (Hendrycks et al., 2021), OlympiadBench (He et al., 2024), AMC23 (math ai, 2025), AIME24 (H4, 2025) and AIME25 (OpenCompass, 2025), using a rollout temperature of $1.0$ during training and a validation temperature of $0.7$. Following DAPO (Yu et al., 2026), we adopt asymmetric clipping for Qwen3-1.7B (high 0.25, low 0.20) and Ministral-8B (high 0.22, low 0.20); Qwen2.5-Math models keep symmetric clipping (0.20/0.20). We use DAPO's length-dependent loss aggregation across all models. Experimental details are in App.C.3.

### 3.4. Comparison of DS-GRPO with Vanilla GRPO

As shown in Figure 2, DS-GRPO demonstrates remarkable robustness. It consistently enhances Pass@K (for most $K$) by $\approx 4\%$ compared to vanilla GRPO. Crucially, these performance gains are accompanied by a $4\times$ inference speedup. (See Appendix D for additional results).

Figure 3 extends our evaluation to three base models and five mathematical reasoning benchmarks. Our strategy yields substantial improvements, with Pass@1 gains of up to 1.1% and Pass@64 gains of up to 11.4% (on AIME25); the largest gains concentrate on the harder AIME 2024/2025 benchmarks, while easier benchmarks (MATH500, Olympiad-Bench) see smaller but mostly positive improvements (detailed in Appendix E.1). This demonstrates our method's ability to improve RL reasoning while mitigating diversity collapse. Furthermore, it delivers significant efficiency gains: DS-GRPO matches the Pass@64 of vanilla GRPO using only $k = 16$ samples—yielding a nearly $4\times$ inference speedup—while simultaneously pushing the maximum

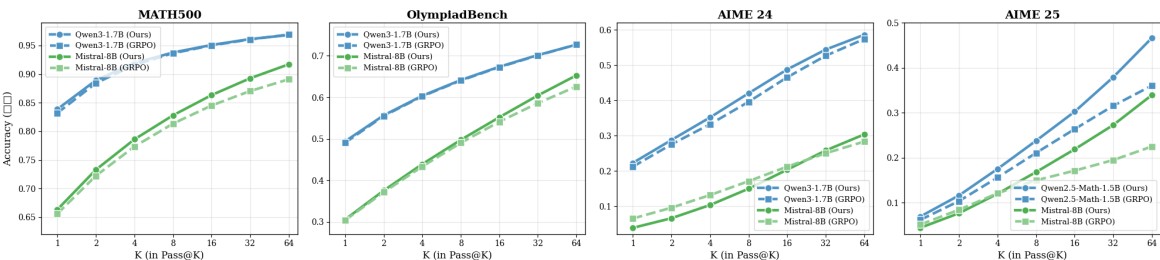

*Figure 3. Pass@K performance after reward modification, compared with vanilla GRPO. X-axis denotes K and y-axis denotes pass rates.*

achievable Pass@K. This uniform uplift underscores our approach's efficacy in enhancing both exploration and diversity.

### 3.5. Ablation Experiments on Hyperparameters

To demonstrate the robustness of our method, we further evaluate performance across different hyperparameters, including sampling temperature, the KL coefficient $\beta_{KL}$, and the reward modification coefficients $\gamma_p$ and $\gamma_n$.

**Temperature and KL Coefficient.** We evaluate the stability of DS-GRPO across varying sampling temperatures and KL coefficients ($\beta_{KL}$). Our results demonstrate consistent improvements over vanilla GRPO: DS-GRPO enhances Pass@K (for all $K$) by $\approx 4\%$ across the temperature range and by $\approx 3.2\%$ across different KL coefficients.

**Reward Modification Coefficient** To isolate the contribution of each component in our reward modification strategy, we conduct an ablation study. We compare the full DS-GRPO algorithm against two specialized variants: *DS-GRPO-Positive*, which only modifies the advantage for correct trajectories, and *DS-GRPO-Negative*, which only modifies the advantage for incorrect trajectories. Their respective advantage modifications are defined as follows:

$$A_i^{DS+} = A_i - \gamma_p \, \log \pi_{\theta_{old}}(y_i \mid x), \quad \text{if } r_i = 1,$$
$$A_i^{DS-} = A_i + \gamma_n \, \log \pi_{\theta_{old}}(y_i \mid x), \quad \text{if } r_i \neq 1.$$

The full DS-GRPO algorithm demonstrates superior performance over both of its individual components (DS-GRPO-Positive and DS-GRPO-Negative) for all $K$. Detailed results and discussion are available in Section E.2. sharpening.

### 3.6. Comparison with Other Methods for Increasing Diversity

We compare DS-GRPO with prior approaches that encourage diversity in reasoning through reward or advantage shaping, either by optimizing Pass@$K$ rate directly(Tang et al., 2025; Walder & Karkhanis, 2026; Chen et al., 2025b) or by applying rank-based penalties (He et al., 2025a). For mathematical reasoning experiments, we use the Qwen2.5-Math-7B(Hui et al., 2024) as the base model. For PKPO and GR-PKPO, we use $K = 4$ which performs best in previous work (Chen et al., 2025b); for rank-based panelty, we sweep

across various configurations. See more result details in Appendix D.

**Pass@K Optimization Methods.** Methods that directly optimize the Pass@$K$ metric use it as a reward signal (Tang et al., 2025; Walder & Karkhanis, 2026; Chen et al., 2025b). However, this approach can assign zero reward to correct solutions, which increases gradient variance and harms training stability. Experimentally, these methods often trade correctness for diversity; for instance, GR-PKPO slightly improves Pass@64 at the cost of Pass@1 and is unstable on the Countdown task (Figure 4). In contrast, DS-GRPO consistently improves Pass@$K$ across all values of $K$.

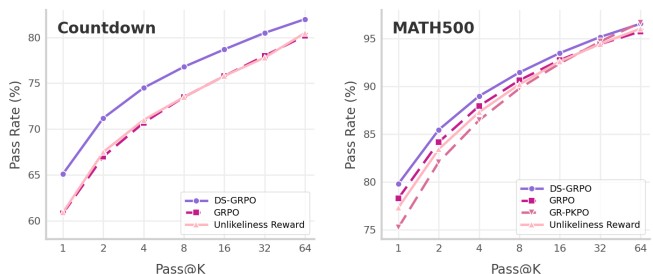

*Figure 4. Comparison with Other Methods*

**Comparison with Unlikeliness Reward Method.** Our work is conceptually similar to methods that reward unlikely solutions, such as the one proposed by He et al. (2025a). However, our approach has key advantages. DS-GRPO is derived from a theoretical framework that guarantees its optimality. More critically, it employs a differentiated reward strategy: it modifies rewards for correct trajectories to boost diversity, while a complementary modification for incorrect trajectories improves correctness. In contrast, methods like that of He et al. (2025a) focus solely on diversity, which can harm correctness. Our experimental results (Figure 4) validate this, showing DS-GRPO's superior performance across all values of $K$.

**Comparison with Other RL Reasoning Methods.** We further compare our approach with a recently proposed method that focuses on improving RL reasoning: CISPO (Chen et al., 2025a). However, in our setup, empirical results demonstrate that our method outperforms CISPO at most $K$ across all four mathematical reasoning datasets, with the largest gains at large $K$. Detailed results are provided in Appendix E.3.

### 3.7. Comparing Differential Approach to Entropy Control Method

Entropy control is a widely adopted strategy for enhancing the diversity and reasoning capabilities of large language models. In Section 2.4, we have theoretically shown that our differential control approach outperforms both baseline methods and those based on direct entropy maximization. This section presents an empirical comparison of our method against these prior entropy control strategies. We demonstrate that our method achieves superior performance compared to both the vanilla GRPO baseline and GRPO augmented with direct entropy maximization.

#### 3.7.1. DS-GRPO OUTPERFORM ENTROPY BASED METHOD

The role of entropy in RL fine-tuning is complex and subject to ongoing debate. While conventional methods employ entropy regularization to prevent policy collapse (Schulman et al., 2017), recent studies suggest that explicitly *minimizing* entropy can, counter-intuitively, boost performance in certain scenarios (Agarwal et al., 2026; Wang et al.). To position our method within this context, we compare it against two direct entropy control baselines: one that adds an **entropy bonus** to encourage exploration, and one that applies an **entropy penalty** to encourage exploitation. The respective optimization objectives are:

> **Entropy Based Method**
>
> $$\mathcal{J}_{\text{ent+}} = \mathcal{J}_{\text{GRPO}}(\theta) - \eta_+ \sum_y \pi_\theta(y|x) \log(\pi_\theta(y|x)),$$
> $$\mathcal{J}_{\text{ent-}} = \mathcal{J}_{\text{GRPO}}(\theta) + \eta_- \sum_y \pi_\theta(y|x) \log(\pi_\theta(y|x)),$$

As illustrated in Fig. 5, our method consistently outperforms both the entropy bonus and penalty approaches, regardless of the direction of the regularization. This suggests that a simple, global adjustment to entropy is less effective than our differentiated reward strategy.

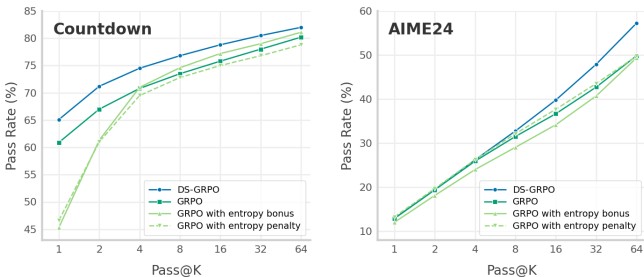

*Figure 5. Comparison with Entropy Control Method*

#### 3.7.2. DEEPER DISCUSSION ON ENTROPY CONTROL

Our experiments (Fig. 5) reveal a critical insight: the effectiveness of global entropy control is highly task-dependent. Specifically, an entropy bonus improves performance over vanilla GRPO on the Countdown task (at large $K$) but hinders it on math reasoning benchmarks. Conversely, an entropy penalty benefits math reasoning while degrading performance on Countdown. *Can we explain these differing trends?*

**A Principle for Task-Aware Entropy Control.** Our theoretical analysis in Section 2.4 reveals a fundamental conflict: global entropy bonuses improve diversity but degrade correctness, while penalties enhance precision but reduce variety. Therefore, effective entropy control must be task-aware: aligning the regularization direction (bonus vs. penalty) with the specific diversity-accuracy demands of the downstream application.

To quantify task-specific diversity requirements, we define Solution Multiplicity (Eq. 2). We assess this metric across four tasks and correlate it with Pass@8 variations induced by entropy bonuses (Table 1; see Appendix C.4). We find that for tasks with abundant unique solutions, the gains from diversity outweigh the trade-off in individual correctness. This motivates our task-aware control principle: employ an entropy bonus for tasks with high Solution Multiplicity, and an entropy penalty for those with low Solution Multiplicity.

**Differential Control for Correct and Incorrect Trajectories.** The underlying mechanism of DS-GRPO is similar to a form of *differentiated entropy control*.

> **Differential Entropy Control**
>
> An objective function representing this principle can be formulated as:
> $$\mathcal{J}_{\text{DS-En}} = \mathcal{J}_{\text{GRPO}} - \eta_p \sum_{y:r(y)>0} \pi_\theta(y|x) \log \pi_\theta(\tau|x)$$
> $$+ \eta_n \sum_{y:r(y)\leq 0} \pi_\theta(y|x) \log \pi_\theta(y|x).$$

By selectively increasing entropy only for positive samples, we attain the full diversity benefits of traditional entropy regularization, as we are only concerned with diversity among correct solutions. Concurrently, decreasing entropy for negative samples reinforces correctness. This targeted approach enables simultaneous gains in both correctness (Pass@1) and diversity (Pass@K), offering a more robust and principled method for model fine-tuning across different tasks. The experimental results for differential entropy control is presented in Appendix F.2

## 4. Related Work

Reinforcement fine-tuning for LLMs is a nuanced process that involves several widely studied ingredients, including surrogate objectives (Tang et al., 2025; Tajwar et al., 2026), policy-gradient baselines (Ahmadian et al., 2024; Li et al., 2023; Zeng et al., 2025), entropy mechanisms (Agarwal et al., 2026; Gao et al., 2025) and so on. While RLVR

effectively enhances reasoning, it frequently induces diversity collapse, degrading generation quality at large sample sizes (Dang et al., 2025; Chen et al., 2026b). Existing countermeasures, ranging from algorithmic adjustments to diversity-aware reward designs (Yu et al., 2026; Walder & Karkhanis, 2026; He et al., 2025a), often force a trade-off between diversity and correctness. Furthermore, the optimal strategy for entropy regulation remains debated: some advocate for maximization to aid exploration (He et al., 2025b), while others observe benefits from minimization (Agarwal et al., 2026). Unlike these uniform strategies, we propose a conditional approach—incentivizing entropy for correct trajectories while suppressing it for incorrect ones. We provide a detailed discussion of these related works in Appendix B.

## 5. Conclusion

In this work, we conduct a formal, first-principles analysis of diversity collapse, from which we derive a novel method to enhance policy diversity. We empirically demonstrate that our method outperforms existing approaches and theoretically prove its optimality. Our analysis also clarifies the nuanced, task-dependent role of entropy in fine-tuning, leading to a principled control strategy that simultaneously improves both correctness (Pass@1) and diversity (Pass@K). A formal theoretical analysis of our entropy principle and the nuanced effects of entropy it reveals is left as a promising direction for future research.

## Impact Statement

This paper presents work whose goal is to advance the field of machine learning. There are many potential societal consequences of our work, none of which we feel must be specifically highlighted here.

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

# A. Additional Related Work

**Mitigating Diversity Collapse in RL for Reasoning.** Reinforcement Learning with Verifiable Rewards (RLVR) has become a dominant paradigm for enhancing reasoning capabilities in LLMs (Guo et al., 2025; Jaech et al., 2024). While RLVR effectively "sharpens" the policy toward high-quality sequences (Huang et al., 2025; Pang et al., 2023), recent studies indicate that it often induces *diversity collapse* (Dang et al., 2025; Chen et al., 2026b). This phenomenon manifests as a degradation in Pass@$k$ performance for large $k$, suggesting that the model's generation distribution becomes overly concentrated around a few solutions (Wu et al.).

Various strategies have been proposed to mitigate this issue. From an algorithmic perspective, Yu et al. (2026) introduce a clip-higher strategy and remove KL divergence penalties in GRPO, while He et al. (2025b) utilize adaptive entropy regularization. Zhu et al. (2026) observe that while positive samples sharpen the distribution, penalizing negative samples helps preserve diversity. In terms of reward design, explicitly diversity-aware objectives have been explored, such as optimizing directly for Pass@$k$ (Walder & Karkhanis, 2026; Chen et al., 2025b), incorporating rank-based penalties (He et al., 2025a), or adding entropy terms to advantage estimation (Cui et al., 2025). However, these methods often force a trade-off: enhancing diversity (Pass@$k$) frequently comes at the expense of correctness (Pass@1) or stability. Unlike prior works that apply global constraints, we propose a differential reward structure that fundamentally reconciles this trade-off. While He et al. (2025a) also employ differential rewards, our approach demonstrates greater robustness across diverse tasks, particularly in mathematical reasoning, by rigorously distinguishing between the entropy dynamics of correct and incorrect trajectories.

**Controlling Distribution Entropy in RLVR.** Policy entropy serves as a key indicator of exploration capability (Cheng et al., 2026). However, the optimal strategy for entropy regulation remains a subject of debate. One line of work advocates for entropy maximization to encourage exploration, employing techniques such as entropy bonuses (Yu et al., 2026; He et al., 2025b) or selective training on high-entropy tokens (Wang et al., 2026). Conversely, other studies report that entropy minimization is essential for improving Pass@1 accuracy (Agarwal et al., 2026; Gao et al., 2025), or suggest scheduling entropy reduction over the course of training (Wang et al.). These contradictory findings highlight the limitations of uniform entropy control. Our work addresses this ambiguity by interpreting our method as a form of *conditional* entropy control. In contrast to global maximization or minimization, we incentivize entropy for correct trajectories (to maintain diversity) while penalizing it for incorrect ones (to eliminate errors). We provide both theoretical and empirical evidence that this targeted approach outperforms uniform strategies.

# B. Proof of Theorems

## B.1. Proof of Proposition 6 and Proposition 7

> **Lemma 5.** *The solution to the KL-regularized optimization problem:*
>
> $$\pi_{\beta_{ent}}^* = \arg\max_{\pi} \left\{ \mathbb{E}_{\tau \sim \pi} r(\tau) - \beta_{ent} \cdot \mathbb{D}_{KL}(\pi || \pi_{base}) \right\}$$
>
> *has the following closed-form expression for a trajectory $\tau$:*
>
> $$\pi_{\beta_{ent}}^*(\tau) = \frac{\left[ \prod_{h=1}^{H} \pi_{base,h}(a_h \mid s_h) \right] \exp\left( \frac{1}{\beta_{ent}} r(\tau) \right)}{\sum_{\tau': s_1' = s} \left[ \prod_{h=1}^{H} \pi_{base,h}(a_h' \mid s_h') \right] \exp\left( \frac{1}{\beta_{ent}} r(\tau') \right)},$$
>
> *where the summation in the denominator is over all valid trajectories $\tau'$ starting from the initial state $s$.*

*Proof.* The optimization problem can be written as finding a probability distribution $\pi(\tau)$ over trajectories that solves:

$$\max_{\pi} \sum_{\tau} \pi(\tau) r(\tau) - \beta_{ent} \sum_{\tau} \pi(\tau) \ln\left( \frac{\pi(\tau)}{\pi_{base}(\tau)} \right), \quad \text{subject to} \quad \sum_{\tau} \pi(\tau) = 1.$$

We introduce a Lagrange multiplier $\mu$ for the probability constraint and form the Lagrangian $\mathcal{L}(\pi, \mu)$:

$$\mathcal{L}(\pi, \mu) = \sum_{\tau} \pi(\tau) r(\tau) - \beta_{ent} \sum_{\tau} \pi(\tau) \left[ \ln(\pi(\tau)) - \ln(\pi_{base}(\tau)) \right] - \mu \left( \sum_{\tau} \pi(\tau) - 1 \right).$$

To find the optimal policy, we take the partial derivative of $\mathcal{L}$ with respect to $\pi(\tau)$ and set it to zero:

$$\frac{\partial \mathcal{L}}{\partial \pi(\tau)} = r(\tau) - \beta_{ent} \left( \ln\left( \frac{\pi(\tau)}{\pi_{base}(\tau)} \right) + 1 \right) - \mu = 0.$$

Solving for $\pi(\tau)$, we obtain:

$$\ln\left( \frac{\pi(\tau)}{\pi_{base}(\tau)} \right) = \frac{r(\tau)}{\beta_{ent}} - 1 - \frac{\mu}{\beta_{ent}}$$
$$\implies \pi(\tau) = \pi_{base}(\tau) \exp\left( \frac{r(\tau)}{\beta_{ent}} - 1 - \frac{\mu}{\beta_{ent}} \right) = \pi_{base}(\tau) \exp\left( \frac{r(\tau)}{\beta_{ent}} \right) \exp\left( -1 - \frac{\mu}{\beta_{ent}} \right).$$

The term $\exp(-1 - \mu/\beta_{ent})$ is a constant determined by the normalization constraint $\sum_{\tau'} \pi(\tau') = 1$. Let the partition function be $\mathcal{Z} = \sum_{\tau'} \pi_{base}(\tau') \exp\left( \frac{r(\tau')}{\beta_{ent}} \right)$. The normalization constant must be $1/\mathcal{Z}$, which gives the solution:

$$\pi_{\beta_{ent}}^*(\tau) = \frac{\pi_{base}(\tau) \exp\left( \frac{1}{\beta_{ent}} r(\tau) \right)}{\mathcal{Z}} = \frac{\pi_{base}(\tau) \exp\left( \frac{1}{\beta_{ent}} r(\tau) \right)}{\sum_{\tau'} \pi_{base}(\tau') \exp\left( \frac{1}{\beta_{ent}} r(\tau') \right)}.$$

By substituting the definitions $\pi_{base}(\tau) = \prod_{h=1}^{H} \pi_{base,h}(a_h \mid s_h)$, we arrive at the expression stated in the lemma. This completes the proof. $\square$

> **Proposition 6** (Selection Bias). *The probability that a correct trajectory's likelihood increases is monotonically related to its initial probability under the base model. Formally, for any two correct trajectories $\tau_1, \tau_2$ and $\beta_{ent} > 0$, we have*
>
> $$\pi_{base}(\tau_1) \geq \pi_{base}(\tau_2) \implies \mathbb{P}(\pi_{van}(\tau_1) > \pi_{base}(\tau_1)) \geq \mathbb{P}(\pi_{van}(\tau_2) > \pi_{base}(\tau_2)).$$

> **Proposition 7** (Reinforcement bias). *The magnitude of probability gain for a given trajectory is directly proportional to its probability under the base policy. Formally, if the reward update mechanism has access to the complete set of correct trajectories ($r(\tau) = 1$ for all correct trajectories), then for any trajectory $\tau$ and $\beta_{ent} > 0$, we have*
>
> $$\pi_{van}(\tau) - \pi_{base}(\tau) \propto \pi_{base}(\tau).$$

*Proof.* **Part 1: Monotonicity of Likelihood Improvement.** Under the specified reward mechanism, a correct trajectory $\tau$ receives a positive reward if it is not missed in all $N$ independent samples. This occurs with probability $1 - (1 - \pi_{\text{base}}(\tau))^N$. A positive reward ensures that the likelihood of the trajectory increases after fine-tuning. Thus, the probability of improvement is:

$$\mathbb{P}(\pi_{\text{van}}(\tau) > \pi_{\text{base}}(\tau)) = 1 - (1 - \pi_{\text{base}}(\tau))^N.$$

This function is monotonically increasing with respect to $\pi_{\text{base}}(\tau)$ for $\pi_{\text{base}}(\tau) \in [0, 1]$. Therefore, if $\pi_{\text{base}}(\tau_1) \geq \pi_{\text{base}}(\tau_2)$, the first claim holds.

**Part 2: Proportionality of Probability Gain.** From the closed-form solution for the optimal policy (as derived in Lemma 5), we have $\pi^*_{\beta_{\text{ent}}}(\tau) = \pi_{\text{base}}(\tau) \exp(r(\tau)/\beta_{\text{ent}})/\mathcal{Z}$, where $\mathcal{Z}$ is the partition function. The change in probability is:

$$\pi^*_{\beta_{\text{ent}}}(\tau) - \pi_{\text{base}}(\tau) = \frac{\pi_{\text{base}}(\tau) \exp(r(\tau)/\beta_{\text{ent}})}{\mathcal{Z}} - \pi_{\text{base}}(\tau) = \pi_{\text{base}}(\tau) \left[ \frac{\exp(r(\tau)/\beta_{\text{ent}}) - \mathcal{Z}}{\mathcal{Z}} \right].$$

The partition function $\mathcal{Z} = \sum_{\tau'} \pi_{\text{base}}(\tau') \exp(r(\tau')/\beta_{\text{ent}})$ is a constant for a given policy $\pi_{\text{base}}$ and reward function $r$. The term in the brackets is therefore constant for all trajectories $\tau$ that share the same reward value (e.g., all correct trajectories). Consequently, the probability gain is directly proportional to the initial probability $\pi_{\text{base}}(\tau)$. $\square$

### B.2. Proof of Theorem 4

> **Lemma 8.** *The solution to the KL-regularized optimization problem with an entropy-based reward modification:*
>
> $$\pi^*_{\beta_{ent}, \gamma_{ent}} = \arg\max_\pi \left\{ \mathbb{E}_{\tau \sim \pi} \left[ r(\tau) - \gamma_{ent} \log(\pi_{base}(\tau)) \right] - \beta_{ent} \cdot \mathbb{D}_{KL}(\pi || \pi_{base}) \right\}$$
>
> *is given by:*
>
> $$\pi^*_{\beta_{ent}, \gamma_{ent}}(\tau) = \frac{[\pi_{base}(\tau)]^{1 - \frac{\gamma_{ent}}{\beta_{ent}}} \exp\left( \frac{1}{\beta_{ent}} r(\tau) \right)}{\sum_{\tau'} [\pi_{base}(\tau')]^{1 - \frac{\gamma_{ent}}{\beta_{ent}}} \exp\left( \frac{1}{\beta_{ent}} r(\tau') \right)},$$
>
> *where the summation in the denominator is over all valid trajectories $\tau'$.*

*Proof.* The objective function can be expanded as:

$$\max_\pi \sum_\tau \pi(\tau) \left( r(\tau) - \gamma_{\text{ent}} \log(\pi_{\text{base}}(\tau)) \right) - \beta_{\text{ent}} \sum_\tau \pi(\tau) \ln\left( \frac{\pi(\tau)}{\pi_{\text{base}}(\tau)} \right),$$

subject to the constraint $\sum_\tau \pi(\tau) = 1$. We form the Lagrangian $\mathcal{L}(\pi, \mu)$:

$$\mathcal{L}(\pi, \mu)$$
$$= \sum_\tau \pi(\tau) \left( r(\tau) - \gamma_{\text{ent}} \log(\pi_{\text{base}}(\tau)) \right) - \beta_{\text{ent}} \sum_\tau \pi(\tau) \left( \ln(\pi(\tau)) - \ln(\pi_{\text{base}}(\tau)) \right) - \mu \left( \sum_\tau \pi(\tau) - 1 \right).$$

Setting the partial derivative with respect to $\pi(\tau)$ to zero yields:

$$\frac{\partial \mathcal{L}}{\partial \pi(\tau)} = r(\tau) - \gamma_{\text{ent}} \log(\pi_{\text{base}}(\tau)) - \beta_{\text{ent}} \left( \ln\left( \frac{\pi(\tau)}{\pi_{\text{base}}(\tau)} \right) + 1 \right) - \mu = 0.$$

Solving for $\pi(\tau)$:

$$\ln\left(\frac{\pi(\tau)}{\pi_{\text{base}}(\tau)}\right) = \frac{r(\tau)}{\beta_{\text{ent}}} - \frac{\gamma_{\text{ent}}}{\beta_{\text{ent}}}\log(\pi_{\text{base}}(\tau)) - 1 - \frac{\mu}{\beta_{\text{ent}}}$$

$$\implies \pi(\tau) = \pi_{\text{base}}(\tau)\exp\left(\frac{r(\tau)}{\beta_{\text{ent}}} - \frac{\gamma_{\text{ent}}}{\beta_{\text{ent}}}\log(\pi_{\text{base}}(\tau)) - 1 - \frac{\mu}{\beta_{\text{ent}}}\right)$$

$$= \pi_{\text{base}}(\tau)\cdot(\exp(\log(\pi_{\text{base}}(\tau))))^{-\frac{\gamma_{\text{ent}}}{\beta_{\text{ent}}}}\cdot\exp\left(\frac{r(\tau)}{\beta_{\text{ent}}}\right)\cdot\exp\left(-1 - \frac{\mu}{\beta_{\text{ent}}}\right)$$

$$= [\pi_{\text{base}}(\tau)]^{1-\frac{\gamma_{\text{ent}}}{\beta_{\text{ent}}}}\exp\left(\frac{r(\tau)}{\beta_{\text{ent}}}\right)\exp\left(-1 - \frac{\mu}{\beta_{\text{ent}}}\right).$$

The term $\exp(-1 - \mu/\beta_{\text{ent}})$ is a normalization constant. By enforcing the constraint $\sum_{\tau'}\pi(\tau') = 1$, we find that this constant is the reciprocal of the partition function $\mathcal{Z} = \sum_{\tau'}[\pi_{\text{base}}(\tau')]^{1-\frac{\gamma_{\text{ent}}}{\beta_{\text{ent}}}}\exp(r(\tau')/\beta_{\text{ent}})$. This gives the final solution stated in the lemma. $\qquad\square$

**Lemma 9.** *Consider the reward function $r_{DS}(\tau)$ which modifies the reward based on trajectory correctness, defined by a set of correct trajectories $\mathcal{C}$:*

$$r_{DS}(\tau) = \begin{cases} r(\tau) - \gamma_{DS}\log(\pi_{base}(\tau)) & \text{if } \tau \in \mathcal{C} \\ r(\tau) & \text{if } \tau \notin \mathcal{C} \end{cases}$$

*The solution to the KL-regularized optimization problem $\pi_{DS} = \arg\max_\pi\{\mathbb{E}_{\tau\sim\pi}[r_{DS}(\tau)] - \beta_{DS}\cdot\mathbb{D}_{KL}(\pi||\pi_{base})\}$ is given by:*

$$\pi_{DS}(\tau) = \frac{1}{\mathcal{Z}} \times \begin{cases} [\pi_{base}(\tau)]^{1-\frac{\gamma_{DS}}{\beta_{DS}}}\exp\left(\frac{1}{\beta_{DS}}r(\tau)\right) & \text{if } \tau \in \mathcal{C} \\ [\pi_{base}(\tau)]\cdot\exp\left(\frac{1}{\beta_{DS}}r(\tau)\right) & \text{if } \tau \notin \mathcal{C} \end{cases}$$

*where $\mathcal{Z}$ is the partition function ensuring normalization.*

*Proof.* The objective function is maximized subject to $\sum_\tau\pi(\tau) = 1$. The Lagrangian is:

$$\mathcal{L}(\pi,\mu) = \sum_{\tau\in\mathcal{C}}\pi(\tau)\left(r(\tau) - \gamma_{DS}\log(\pi_{\text{base}}(\tau))\right) + \sum_{\tau\notin\mathcal{C}}\pi(\tau)r(\tau)$$

$$- \beta_{DS}\sum_\tau\pi(\tau)\left(\ln(\pi(\tau)) - \ln(\pi_{\text{base}}(\tau))\right) - \mu\left(\sum_\tau\pi(\tau) - 1\right).$$

We take the partial derivative with respect to $\pi(\tau)$ for each case and set it to zero.

For a correct trajectory, $\tau \in \mathcal{C}$:

$$\frac{\partial\mathcal{L}}{\partial\pi(\tau)} = r(\tau) - \gamma_{DS}\log(\pi_{\text{base}}(\tau)) - \beta_{DS}\left(\ln\left(\frac{\pi(\tau)}{\pi_{\text{base}}(\tau)}\right) + 1\right) - \mu = 0.$$

Solving for $\pi(\tau)$ yields: $\pi(\tau) \propto [\pi_{\text{base}}(\tau)]^{1-\frac{\gamma_{DS}}{\beta_{DS}}}\exp\left(\frac{r(\tau)}{\beta_{DS}}\right)$.

For an incorrect trajectory, $\tau \notin \mathcal{C}$:

$$\frac{\partial\mathcal{L}}{\partial\pi(\tau)} = r(\tau) - \beta_{DS}\cdot\left(\ln\left(\frac{\pi(\tau)}{\pi_{\text{base}}(\tau)}\right) + 1\right) - \mu = 0.$$

Solving for $\pi(\tau)$ yields: $\pi(\tau) \propto [\pi_{\text{base}}(\tau)]\cdot\exp\left(\frac{r(\tau)}{\beta_{\text{ent}}}\right)$.

Combining these results, the unnormalized solution $\tilde{\pi}(\tau)$ is:

$$\tilde{\pi}(\tau) = \begin{cases} [\pi_{\text{base}}(\tau)]^{1-\frac{\gamma_{DS}}{\beta_{\text{ent}}}}\exp\left(\frac{1}{\beta_{\text{ent}}}r(\tau)\right) & \text{if } \tau \in \mathcal{C} \\ [\pi_{\text{base}}(\tau)]\cdot\exp\left(\frac{1}{\beta_{\text{ent}}}r(\tau)\right) & \text{if } \tau \notin \mathcal{C} \end{cases}$$

The final solution $\pi_{\mathrm{DS}}$ is obtained by normalizing $\tilde{\pi}(\tau)$ with the partition function $\mathcal{Z} = \sum_{\tau'} \tilde{\pi}(\tau')$, which gives the expression stated in the lemma. $\qquad\square$

**Lemma 10** (Correctness under Reverse KL Constraint). *When $K_\rho(\pi, \pi_{base}) = \mathbb{D}_{KL}(\pi_{base}\|\pi)$, for any $\gamma_{ent} \geq 0, \beta_{ent} > 0$ such that $\mathbb{D}_{KL}(\pi_{base}\|\pi_{ent}) \leq \kappa$, there exist $\gamma_{DS} \geq 0$ and $\beta_{DS} > 0$ such that $\mathbb{D}_{KL}(\pi_{base}\|\pi_{DS}) \leq \kappa$ and $C(\pi_{DS}) \geq C(\pi_{ent})$, and $\sigma(\pi_{DS}) \geq \sigma(\pi_{ent})$.*

*Proof.* To simplify the notation, we define the following sums over trajectory probabilities, where $\mathcal{C}$ is the set of correct trajectories:

$$b_x = \sum_{\tau \in \mathcal{C}} [\pi_{\mathrm{base}}(\tau)]^{1-x}, \quad B_x = \sum_{\tau} [\pi_{\mathrm{base}}(\tau)]^{1-x}, \quad p_c = \sum_{\tau \in \mathcal{C}} \pi_{\mathrm{base}}(\tau).$$

The proof proceeds in three steps: we first find a functional relationship between correctness $C$ and the KL divergence for each policy, and then compare them.

**Step 1: Analyze the Entropy-Maximization Policy ($\pi_{\mathbf{ent}}$).** The correctness is the total probability mass on correct trajectories:

$$C(\pi_{\mathrm{ent}}) = \frac{\sum_{\tau \in \mathcal{C}} [\pi_{\mathrm{base}}(\tau)]^{1-\frac{\gamma_{\mathrm{ent}}}{\beta_{\mathrm{ent}}}} \exp(1/\beta_{\mathrm{ent}})}{\sum_{\tau \in \mathcal{C}} [\pi_{\mathrm{base}}(\tau)]^{1-\frac{\gamma_{\mathrm{ent}}}{\beta_{\mathrm{ent}}}} \exp(1/\beta_{\mathrm{ent}}) + \sum_{\tau \notin \mathcal{C}} [\pi_{\mathrm{base}}(\tau)]^{1-\frac{\gamma_{\mathrm{ent}}}{\beta_{\mathrm{ent}}}}} = \frac{b_{\frac{\gamma_{\mathrm{ent}}}{\beta_{\mathrm{ent}}}} e^{1/\beta_{\mathrm{ent}}}}{b_{\frac{\gamma_{\mathrm{ent}}}{\beta_{\mathrm{ent}}}} e^{1/\beta_{\mathrm{ent}}} + \left(B_{\frac{\gamma_{\mathrm{ent}}}{\beta_{\mathrm{ent}}}} - b_{\frac{\gamma_{\mathrm{ent}}}{\beta_{\mathrm{ent}}}}\right)}.$$

Solving for $e^{1/\beta_{\mathrm{ent}}}$ gives: $e^{1/\beta_{\mathrm{ent}}} = \frac{B_{\frac{\gamma_{\mathrm{ent}}}{\beta_{\mathrm{ent}}}} - b_{\frac{\gamma_{\mathrm{ent}}}{\beta_{\mathrm{ent}}}}}{b_{\frac{\gamma_{\mathrm{ent}}}{\beta_{\mathrm{ent}}}}} \left(\frac{C(\pi_{\mathrm{ent}})}{1-C(\pi_{\mathrm{ent}})}\right)$. The reverse KL divergence is $\mathbb{D}_{\mathrm{KL}}(\pi_{\mathrm{base}}\|\pi_{\mathrm{ent}}) = \sum_{\tau} \pi_{\mathrm{base}}(\tau) \ln(\pi_{\mathrm{base}}(\tau)/\pi_{\mathrm{ent}}(\tau))$. Substituting the policy definition:

$$\mathbb{D}_{\mathrm{KL}}(\pi_{\mathrm{base}}\|\pi_{\mathrm{ent}}) = \frac{\gamma_{\mathrm{ent}}}{\beta_{\mathrm{ent}}} \sum_{\tau} \pi_{\mathrm{base}} \ln \pi_{\mathrm{base}} - \frac{1}{\beta_{\mathrm{ent}}} \sum_{\tau} \pi_{\mathrm{base}} \cdot r(\tau) + \ln\left(b_{\frac{\gamma_{\mathrm{ent}}}{\beta_{\mathrm{ent}}}} e^{1/\beta_{\mathrm{ent}}} + B_{\frac{\gamma_{\mathrm{ent}}}{\beta_{\mathrm{ent}}}} - b_{\frac{\gamma_{\mathrm{ent}}}{\beta_{\mathrm{ent}}}}\right)$$

$$= \frac{\gamma_{\mathrm{ent}}}{\beta_{\mathrm{ent}}} \sum_{\tau} \pi_{\mathrm{base}} \ln \pi_{\mathrm{base}} - \frac{p_c}{\beta_{\mathrm{ent}}} + \ln\left(\frac{b_{\frac{\gamma_{\mathrm{ent}}}{\beta_{\mathrm{ent}}}} e^{1/\beta_{\mathrm{ent}}}}{C(\pi_{\mathrm{ent}})}\right).$$

Substituting the expression for $1/\beta_{\mathrm{ent}}$ leads to a relationship between divergence and correctness:

$$\mathbb{D}_{\mathrm{KL}}(\pi_{\mathrm{base}}\|\pi_{\mathrm{ent}}) = \frac{\gamma_{\mathrm{ent}}}{\beta_{\mathrm{ent}}} \sum_{\tau} \pi_{\mathrm{base}} \ln \pi_{\mathrm{base}} + p_c \ln b_{\frac{\gamma_{\mathrm{ent}}}{\beta_{\mathrm{ent}}}} + (1-p_c) \ln(B_{\frac{\gamma_{\mathrm{ent}}}{\beta_{\mathrm{ent}}}} - b_{\frac{\gamma_{\mathrm{ent}}}{\beta_{\mathrm{ent}}}})$$
$$- [p_c \ln C(\pi_{\mathrm{ent}}) + (1-p_c) \ln(1 - C(\pi_{\mathrm{ent}}))].$$

**Step 2: Analyze Differential Policy ($\pi_{\mathbf{DS}}$).** Similarly, the correctness is:

$$C(\pi_{\mathrm{DS}}) = \frac{b_{\frac{\gamma_{\mathrm{DS}}}{\beta_{\mathrm{DS}}}} e^{1/\beta_{\mathrm{DS}}}}{b_{\frac{\gamma_{\mathrm{DS}}}{\beta_{\mathrm{DS}}}} e^{1/\beta_{\mathrm{DS}}} + (1-p_c)}.$$

The reverse KL divergence, after a similar derivation, is:

$$\mathbb{D}_{\mathrm{KL}}(\pi_{\mathrm{base}}\|\pi_{\mathrm{DS}}) = \frac{\gamma_{\mathrm{DS}}}{\beta_{\mathrm{DS}}} \sum_{\tau \in \mathcal{C}} \pi_{\mathrm{base}} \ln \pi_{\mathrm{base}} + p_c \ln b_{\frac{\gamma_{\mathrm{DS}}}{\beta_{\mathrm{DS}}}} + (1-p_c) \ln(1 - p_c)$$
$$- [p_c \ln C(\pi_{\mathrm{DS}}) + (1-p_c) \ln(1 - C(\pi_{\mathrm{DS}}))]. \tag{5}$$

**Step 3: Compare the Policies.** Our goal is to show that for any $C(\pi_{\mathrm{ent}})$, we can choose parameters for our method to achieve $C(\pi_{\mathrm{DS}}) = C(\pi_{\mathrm{ent}})$ with a smaller or equal KL divergence. Let's choose $\gamma_{\mathrm{DS}} = \frac{\beta_{\mathrm{DS}}}{\beta_{\mathrm{ent}}} \gamma_{\mathrm{ent}}$, and set $C(\pi_{\mathrm{DS}}) = C(\pi_{\mathrm{ent}}) = C$. Equivalently, we assume that

$$\frac{\gamma_{\mathrm{DS}}}{\beta_{\mathrm{DS}}} = \frac{\gamma_{\mathrm{ent}}}{\beta_{\mathrm{ent}}} = \widetilde{\gamma}.$$

Then the KL divergence for our method becomes:

$$\mathbb{D}_{\text{KL}}(\pi_{\text{base}}\|\pi_{\text{DS}}) = \widetilde{\gamma} \sum_{\tau \in \mathcal{C}} \pi_{\text{base}} \ln \pi_{\text{base}} + p_c \ln b_{\gamma_{\text{ent}}} + (1 - p_c) \ln(1 - p_c) - H(C_{\text{ent}}),$$

where $H(C) = -[p_c \ln C + (1 - p_c) \ln(1 - C)]$. For the entropy method, the KL is:

$$\mathbb{D}_{\text{KL}}(\pi_{\text{base}}\|\pi_{\text{ent}}) = \widetilde{\gamma} \sum_{\tau \in \mathcal{C}} \pi_{\text{base}} \ln \pi_{\text{base}} + \widetilde{\gamma} \sum_{\tau \notin \mathcal{C}} \pi_{\text{base}} \ln \pi_{\text{base}} + p_c \ln b_{\widetilde{\gamma}} + (1 - p_c) \ln(B_{\widetilde{\gamma}} - b_{\widetilde{\gamma}}) - H(C_{\text{ent}}).$$

The difference is $\mathbb{D}_{\text{KL}}(\pi_{\text{base}}\|\pi_{\text{ent}}) - \mathbb{D}_{\text{KL}}(\text{ours}) = \widetilde{\gamma} \sum_{\tau \notin \mathcal{C}} \pi_{\text{base}} \ln \pi_{\text{base}} + (1 - p_c) \ln(B_{\widetilde{\gamma}} - b_{\widetilde{\gamma}}) - (1 - p_c) \ln(1 - p_c)$. By Jensen's inequality on the concave function $\ln(\cdot)$:

$$\sum_{\tau \notin \mathcal{C}} \frac{\pi_{\text{base}}(\tau)}{1 - p_c} \ln \left( [\pi_{\text{base}}(\tau)]^{-\widetilde{\gamma}} \right) \leq \ln \left( \sum_{\tau \notin \mathcal{C}} \frac{\pi_{\text{base}}(\tau)}{1 - p_c} [\pi_{\text{base}}(\tau)]^{-\widetilde{\gamma}} \right) = \ln \left( \frac{B_{\widetilde{\gamma}} - b_{\widetilde{\gamma}}}{1 - p_c} \right)$$

Multiplying by $-(1 - p_c)$ gives:

$$\widetilde{\gamma} \sum_{\tau \notin \mathcal{C}} \pi_{\text{base}}(\tau) \ln(\pi_{\text{base}}(\tau)) \geq -(1 - p_c) \ln \left( \frac{B_{\widetilde{\gamma}} - b_{\widetilde{\gamma}}}{1 - p_c} \right) = -(1 - p_c) \left[ \ln(B_{\widetilde{\gamma}} - b_{\widetilde{\gamma}}) - \ln(1 - p_c) \right].$$

Furthermore, assuming the ratio equality $\frac{\gamma_{\text{DS}}}{\beta_{\text{DS}}} = \frac{\gamma_{\text{ent}}}{\beta_{\text{ent}}} = \widetilde{\gamma}$, the policy $\pi_{\text{DS}}$ takes the form:

$$\pi_{\text{DS}}(\tau) = \frac{1}{\mathcal{Z}} \begin{cases} [\pi_{\text{base}}(\tau)]^{1-\widetilde{\gamma}} \exp \left( \frac{1}{\beta_{\text{DS}}} r(\tau) \right) & \text{if } \tau \in \mathcal{C}, \\ [\pi_{\text{base}}(\tau)] \cdot \exp \left( \frac{1}{\beta_{\text{DS}}} r(\tau) \right) & \text{if } \tau \notin \mathcal{C}, \end{cases}$$

where $\mathcal{Z}$ is the partition function. Similarly, the entropy-regularized policy is given by:

$$\pi_{\text{ent}}^*(\tau) = \frac{[\pi_{\text{base}}(\tau)]^{1-\widetilde{\gamma}} \exp \left( \frac{1}{\beta_{\text{ent}}} r(\tau) \right)}{\sum_{\tau'} [\pi_{\text{base}}(\tau')]^{1-\widetilde{\gamma}} \exp \left( \frac{1}{\beta_{\text{ent}}} r(\tau') \right)}.$$

This yields the equivalence $\sigma(\pi_{\text{DS}}) = \sigma(\pi_{\text{ent}})$, which concludes the proof. □

---

**Lemma 11** (Correctness under Forward KL Constraint). *When $K_\rho(\pi, \pi_{base}) = \mathbb{D}_{KL}(\pi\|\pi_{base})$, for any $\gamma_{ent} \geq 0, \beta_{ent} > 0$ such that $\mathbb{D}_{KL}(\pi_{ent}\|\pi_{base}) \leq \kappa$, there exist $\gamma_{DS} \geq 0$ and $\beta_{DS} > 0$ such that $\mathbb{D}_{KL}(\pi_{DS}\|\pi_{base}) \leq \kappa$ and $C(\pi_{DS}) \geq C(\pi_{ent})$, and $\sigma(\pi_{DS}) \geq \sigma(\pi_{ent})$.*

---

*Proof.* The proof proceeds in three steps: we first find a functional relationship between correctness $C$ and the KL divergence for each policy, and then compare them.

**Step 1: Analyze the Entropy-Maximization Policy ($\pi_{\text{ent}}$).** The correctness of $\pi_{ent}$ is the total probability mass on correct trajectories:

$$C(\pi_{\text{ent}}) = \frac{\sum_{\tau \in \mathcal{C}} [\pi_{\text{base}}(\tau)]^{1 - \frac{\gamma_{\text{ent}}}{\beta_{\text{ent}}}} \exp(1/\beta_{\text{ent}})}{\sum_{\tau \in \mathcal{T}} [\pi_{\text{base}}(\tau)]^{1 - \frac{\gamma_{\text{ent}}}{\beta_{\text{ent}}}} \exp(r(\tau)/\beta_{\text{ent}})} = \frac{b_{\frac{\gamma_{\text{ent}}}{\beta_{\text{ent}}}} e^{1/\beta_{\text{ent}}}}{b_{\frac{\gamma_{\text{ent}}}{\beta_{\text{ent}}}} e^{1/\beta_{\text{ent}}} + (B_{\frac{\gamma_{\text{ent}}}{\beta_{\text{ent}}}} - b_{\frac{\gamma_{\text{ent}}}{\beta_{\text{ent}}}})}. \tag{6}$$

Solving for $e^{1/\beta_{\text{ent}}}$ yields: $e^{1/\beta_{\text{ent}}} = \frac{B_{\frac{\gamma_{\text{ent}}}{\beta_{\text{ent}}}} - b_{\frac{\gamma_{\text{ent}}}{\beta_{\text{ent}}}}}{b_{\frac{\gamma_{\text{ent}}}{\beta_{\text{ent}}}}} \left( \frac{C(\pi_{\text{ent}})}{1 - C(\pi_{\text{ent}})} \right).$

The reverse KL divergence $\mathbb{D}_{\mathrm{KL}}(\pi_{\mathrm{ent}}\|\pi_{\mathrm{base}})$ can be expressed as a function of $C(\pi_{\mathrm{ent}})$. Following the derivation previously, we arrive at:

$$
\begin{aligned}
\mathbb{D}_{\mathrm{KL}}(\pi_{\mathrm{ent}}\|\pi_{\mathrm{base}}) = & -\frac{\frac{\gamma_{\mathrm{ent}}}{\beta_{\mathrm{ent}}}}{b_{\frac{\gamma_{\mathrm{ent}}}{\beta_{\mathrm{ent}}}}} \sum_{\tau \in \mathcal{C}} \pi_{\mathrm{base}}(\tau)^{1-\frac{\gamma_{\mathrm{ent}}}{\beta_{\mathrm{ent}}}} \ln(\pi_{\mathrm{base}}(\tau)) \cdot C(\pi_{\mathrm{ent}}) \\
& - \frac{\frac{\gamma_{\mathrm{ent}}}{\beta_{\mathrm{ent}}}}{B_{\frac{\gamma_{\mathrm{ent}}}{\beta_{\mathrm{ent}}}} - b_{\frac{\gamma_{\mathrm{ent}}}{\beta_{\mathrm{ent}}}}} \sum_{\tau \notin \mathcal{C}} \pi_{\mathrm{base}}(\tau)^{1-\frac{\gamma_{\mathrm{ent}}}{\beta_{\mathrm{ent}}}} \ln(\pi_{\mathrm{base}}(\tau)) \cdot (1 - C(\pi_{\mathrm{ent}})) \\
& + (1 - C(\pi_{\mathrm{ent}})) \ln\left(\frac{b_{\frac{\gamma_{\mathrm{ent}}}{\beta_{\mathrm{ent}}}}}{B_{\frac{\gamma_{\mathrm{ent}}}{\beta_{\mathrm{ent}}}} - b_{\frac{\gamma_{\mathrm{ent}}}{\beta_{\mathrm{ent}}}}}\right) - \ln b_{\frac{\gamma_{\mathrm{ent}}}{\beta_{\mathrm{ent}}}} + H(C(\pi_{\mathrm{ent}})),
\end{aligned}
\tag{7}
$$

where $H(C) = C \ln C + (1-C) \ln(1-C)$ is the binary entropy function.

**Step 2: Analyze Differential Policy ($\pi_{\mathrm{DS}}$).** Similarly, the correctness for our policy is given by:

$$
C(\pi_{\mathrm{DS}}) = \frac{b_{\frac{\gamma_{\mathrm{DS}}}{\beta_{\mathrm{DS}}}} e^{1/\beta_{\mathrm{DS}}}}{b_{\frac{\gamma_{\mathrm{DS}}}{\beta_{\mathrm{DS}}}} e^{1/\beta_{\mathrm{DS}}} + (1-p_c)}.
\tag{8}
$$

The corresponding reverse KL divergence as a function of $C(\pi_{\mathrm{DS}})$ is:

$$
\begin{aligned}
\mathbb{D}_{\mathrm{KL}}(\pi_{\mathrm{DS}}\|\pi_{\mathrm{base}}) = & -\frac{\frac{\gamma_{\mathrm{DS}}}{\beta_{\mathrm{DS}}}}{b_{\frac{\gamma_{\mathrm{DS}}}{\beta_{\mathrm{DS}}}}} \sum_{\tau \in \mathcal{C}} \pi_{\mathrm{base}}(\tau)^{1-\frac{\gamma_{\mathrm{DS}}}{\beta_{\mathrm{DS}}}} \ln(\pi_{\mathrm{base}}(\tau)) \cdot C(\pi_{\mathrm{DS}}) \\
& + (1 - C(\pi_{\mathrm{DS}})) \ln\left(\frac{b_{\frac{\gamma_{\mathrm{DS}}}{\beta_{\mathrm{DS}}}}}{1 - p_c}\right) - \ln b_{\frac{\gamma_{\mathrm{DS}}}{\beta_{\mathrm{DS}}}} + H(C(\pi_{\mathrm{DS}})).
\end{aligned}
\tag{9}
$$

**Step 3: Compare the Policies.** Our goal is to show that for any $C(\pi_{\mathrm{ent}})$, we can choose parameters for our method to achieve $C(\pi_{\mathrm{DS}}) = C(\pi_{\mathrm{ent}})$ with a smaller or equal KL divergence. Let's choose $\gamma_{\mathrm{DS}} = \frac{\beta_{\mathrm{DS}}}{\beta_{\mathrm{ent}}}\gamma_{\mathrm{ent}}$, and set $C(\pi_{\mathrm{DS}}) = C(\pi_{\mathrm{ent}}) = C$. Equivalently, we assume that

$$
\frac{\gamma_{\mathrm{DS}}}{\beta_{\mathrm{DS}}} = \frac{\gamma_{\mathrm{ent}}}{\beta_{\mathrm{ent}}} = \widetilde{\gamma}.
$$

Then the KL divergence for our method becomes: According to Jensen's inequality, we have

$$
\frac{\widetilde{\gamma}\sum_{\tau \notin \mathcal{C}} \pi_{\mathrm{base}}(\tau)^{1-\widetilde{\gamma}} \ln(\pi_{\mathrm{base}}(\tau))}{B_{\widetilde{\gamma}} - b_{\widetilde{\gamma}}} \cdot (1-C) \leq (1-C) \cdot \ln\left(\frac{\sum_{\tau \notin \mathcal{C}} \pi_{\mathrm{base}}(\tau)}{B_{\widetilde{\gamma}} - b_{\widetilde{\gamma}}}\right) = (1-C) \cdot \ln\left(\frac{1 - p_C}{B_{\widetilde{\gamma}} - b_{\widetilde{\gamma}}}\right)
$$

Then in this case we have

$$
\begin{aligned}
\mathbb{D}_{\mathrm{KL}}(\pi_{\mathrm{ent}}\|\pi_{\mathrm{base}}) = & -\frac{\widetilde{\gamma} \cdot \sum_{\tau \in \mathcal{C}} \pi_{\mathrm{base}}(\tau)^{1-\widetilde{\gamma}} \ln(\pi_{\mathrm{base}}(\tau))}{b_{\widetilde{\gamma}}} \cdot C + (1-C)\left[\ln\left(\frac{b_{\widetilde{\gamma}}}{B_{\widetilde{\gamma}} - b_{\widetilde{\gamma}}}\right)\right] - \ln b_{\widetilde{\gamma}} \\
& + C \ln C + (1-C) \ln(1-C) - \frac{\widetilde{\gamma}\sum_{\tau \notin \mathcal{C}} \pi_{\mathrm{base}}(\tau)^{1-\widetilde{\gamma}} \ln(\pi_{\mathrm{base}}(\tau))}{1 - p_C} \cdot (1-C) \\
\geq & -\frac{\widetilde{\gamma}\sum_{\tau \in \mathcal{C}} \pi_{\mathrm{base}}(\tau)^{1-\widetilde{\gamma}} \ln(\pi_{\mathrm{base}}(\tau))}{b_{\widetilde{\gamma}}} \cdot C + (1-C)\left[\ln\left(\frac{b_{\widetilde{\gamma}}}{B_{\widetilde{\gamma}} - b_{\widetilde{\gamma}}}\right)\right] - \ln b_{\widetilde{\gamma}} \\
& + C \ln C + (1-C) \ln(1-C) \\
= & \mathbb{D}_{\mathrm{KL}}(\pi_{\mathrm{DS}}\|\pi_{\mathrm{base}})
\end{aligned}
$$

Therefore, the difference is non-negative: $\mathbb{D}_{\mathrm{KL}}(\pi_{\mathrm{ent}}\|\pi_{\mathrm{base}}) - \mathbb{D}_{\mathrm{KL}}(\pi_{\mathrm{DS}}\|\pi_{\mathrm{base}}) \geq 0$. This means that for any given correctness level $C$, our method (with $\gamma_{\mathrm{DS}} = \gamma_{\mathrm{ent}} \cdot \frac{\beta_{\mathrm{DS}}}{\beta_{\mathrm{ent}}}$) can achieve it with a lower or equal KL-divergence cost. Thus, if the entropy method is constrained by $\mathbb{D}_{\mathrm{KL}} \leq \kappa$, our method can achieve a correctness $C(\pi_{\mathrm{DS}}) \geq C(\pi_{\mathrm{ent}})$ while also satisfying the constraint.

Furthermore, assuming the ratio equality $\frac{\gamma_{\mathrm{DS}}}{\beta_{\mathrm{DS}}} = \frac{\gamma_{\mathrm{ent}}}{\beta_{\mathrm{ent}}} = \widetilde{\gamma}$, the policy $\pi_{\mathrm{DS}}$ takes the form:

$$
\pi_{\mathrm{DS}}(\tau) = \frac{1}{\mathcal{Z}}
\begin{cases}
[\pi_{\mathrm{base}}(\tau)]^{1-\widetilde{\gamma}} \exp\left(\frac{1}{\beta_{\mathrm{DS}}} r(\tau)\right) & \text{if } \tau \in \mathcal{C}, \\
[\pi_{\mathrm{base}}(\tau)] \cdot \exp\left(\frac{1}{\beta_{\mathrm{DS}}} r(\tau)\right) & \text{if } \tau \notin \mathcal{C},
\end{cases}
$$

where $\mathcal{Z}$ is the partition function. Similarly, the entropy-regularized policy is given by:

$$\pi_{\text{ent}}^*(\tau) = \frac{[\pi_{\text{base}}(\tau)]^{1-\widetilde{\gamma}} \exp\left(\frac{1}{\beta_{\text{ent}}} r(\tau)\right)}{\sum_{\tau'} [\pi_{\text{base}}(\tau')]^{1-\widetilde{\gamma}} \exp\left(\frac{1}{\beta_{\text{ent}}} r(\tau')\right)}.$$

This yields the equivalence $\sigma(\pi_{\text{DS}}) = \sigma(\pi_{\text{ent}})$, which concludes the proof. $\qquad\square$

> **Lemma 12** (Correctness under Reverse $\chi^2$ Constraint). *When $K_\rho(\pi, \pi_{base}) = \mathbb{D}_{\chi^2}(\pi_{base}\|\pi)$, for any $\gamma_{ent} \geq 0, \beta_{ent} > 0$ such that $\mathbb{D}_{KL}(\pi_{base}\|\pi_{ent}) \leq \kappa$, there exist $\gamma_{DS} \geq 0$ and $\beta_{DS} > 0$ such that $\mathbb{D}_{\chi^2}(\pi_{base}\|\pi_{DS}) \leq \kappa$ and $C(\pi_{DS}) \geq C(\pi_{ent})$, and $\sigma(\pi_{DS}) \geq \sigma(\pi_{ent})$.*

**Step 1: Analyze the Entropy-Maximization Policy ($\pi_{\text{ent}}$).** The correctness of $\pi_{\text{ent}}$ is the total probability mass on correct trajectories:

$$C(\pi_{\text{ent}}) = \frac{\sum_{\tau \in \mathcal{C}} [\pi_{\text{base}}(\tau)]^{1-\frac{\gamma_{\text{ent}}}{\beta_{\text{ent}}}} \exp(1/\beta_{\text{ent}})}{\sum_{\tau \in \mathcal{T}} [\pi_{\text{base}}(\tau)]^{1-\frac{\gamma_{\text{ent}}}{\beta_{\text{ent}}}} \exp(r(\tau)/\beta_{\text{ent}})} = \frac{b_{\frac{\gamma_{\text{ent}}}{\beta_{\text{ent}}}} e^{1/\beta_{\text{ent}}}}{b_{\frac{\gamma_{\text{ent}}}{\beta_{\text{ent}}}} e^{1/\beta_{\text{ent}}} + (B_{\frac{\gamma_{\text{ent}}}{\beta_{\text{ent}}}} - b_{\frac{\gamma_{\text{ent}}}{\beta_{\text{ent}}}})}. \tag{10}$$

Solving for $e^{1/\beta_{\text{ent}}}$ yields: $e^{1/\beta_{\text{ent}}} = \frac{B_{\frac{\gamma_{\text{ent}}}{\beta_{\text{ent}}}} - b_{\frac{\gamma_{\text{ent}}}{\beta_{\text{ent}}}}}{b_{\frac{\gamma_{\text{ent}}}{\beta_{\text{ent}}}}} \left(\frac{C(\pi_{\text{ent}})}{1-C(\pi_{\text{ent}})}\right).$

The reverse $\chi^2$ divergence $\mathbb{D}_{\chi^2}(\pi_{\text{base}}\|\pi_{\text{ent}})$ can be expressed as a function of $C(\pi_{\text{ent}})$. Following the derivation previously, we arrive at:

$$\mathbb{D}_{\chi^2}(\pi_{\text{base}}\|\pi_{\text{ent}}) = \sum_\tau \pi_{\text{base}}(\tau) \left(\frac{\pi_{\text{ent}}(\tau)}{\pi_{\text{base}}(\tau)} - 1\right)^2 = \sum_\tau \frac{(\pi_{\text{ent}}(\tau))^2}{\pi_{\text{base}}(\tau)} - 1$$

We insert the expression of $\pi_{\text{ent}}$ into the divergence constraints and we can obtain that

$$\mathbb{D}_{\chi^2}(\pi_{\text{base}}\|\pi_{\text{ent}}) = -1 + \frac{e^{\frac{2}{\beta_{\text{ent}}}} \sum_{\tau \in \mathcal{C}} \pi_{\text{base}}(\tau)^{1-2\frac{\gamma_{\text{ent}}}{\beta_{\text{ent}}}} + \sum_{\tau \notin \mathcal{C}} \pi_{\text{base}}(\tau)^{1-2\frac{\gamma_{\text{ent}}}{\beta_{\text{ent}}}}}{\left[B_{\frac{\gamma_{\text{ent}}}{\beta_{\text{ent}}}} - b_{\frac{\gamma_{\text{ent}}}{\beta_{\text{ent}}}} + b_{\frac{\gamma_{\text{ent}}}{\beta_{\text{ent}}}} e^{\frac{1}{\beta_{\text{ent}}}}\right]^2}$$

We then insert Eq. 10 into the expression of divergence and we can obtain that

$$\mathbb{D}_{\chi^2}(\pi_{\text{base}}\|\pi_{\text{ent}}) = -1 + \frac{\sum_{\tau \in \mathcal{C}} \pi_{\text{base}}(\tau)^{1-2\frac{\gamma_{\text{ent}}}{\beta_{\text{ent}}}}}{b_{\frac{\gamma_{\text{ent}}}{\beta_{\text{ent}}}}^2} \cdot C(\pi_{\text{ent}})^2 + (1 - C(\pi_{\text{ent}}))^2 \frac{\sum_{\tau \notin \mathcal{C}} \pi_{\text{base}}(\tau)^{1-2\frac{\gamma_{\text{ent}}}{\beta_{\text{ent}}}}}{\left(B_{\frac{\gamma_{\text{ent}}}{\beta_{\text{ent}}}} - b_{\frac{\gamma_{\text{ent}}}{\beta_{\text{ent}}}}\right)^2}$$

**Step 2: Analyze Differential Policy ($\pi_{\text{DS}}$).** Similarly, the correctness is:

$$C(\pi_{\text{DS}}) = \frac{b_{\frac{\gamma_{\text{DS}}}{\beta_{\text{DS}}}} e^{1/\beta_{\text{DS}}}}{b_{\frac{\gamma_{\text{DS}}}{\beta_{\text{DS}}}} e^{1/\beta_{\text{DS}}} + (1-p_c)}.$$

The reverse $\chi^2$ divergence, after a similar derivation, is:

$$\mathbb{D}_{\chi^2}(\pi_{\text{base}}\|\pi_{\text{DS}}) = -1 + \frac{\sum_{\tau \in \mathcal{C}} \pi_{\text{base}}(\tau)^{1-2\frac{\gamma_{\text{DS}}}{\beta_{\text{DS}}}}}{b_{\frac{\gamma_{\text{DS}}}{\beta_{\text{DS}}}}^2} \cdot C(\pi_{\text{DS}})^2 + (1 - C(\pi_{\text{DS}}))^2 \frac{\sum_{\tau \notin \mathcal{C}} \pi_{\text{base}}(\tau)}{(1-p_c)^2}$$

**Step 3: Compare the Policies.** Our goal is to show that for any $C(\pi_{\text{ent}})$, we can choose parameters for our method to achieve $C(\pi_{\text{DS}}) \geq C(\pi_{\text{ent}})$ with a smaller or equal $\chi^2$ divergence. Let's choose $\gamma_{\text{DS}} = \frac{\beta_{\text{DS}}}{\beta_{\text{ent}}} \gamma_{\text{ent}}$, and set $C(\pi_{\text{DS}}) = C(\pi_{\text{ent}}) = C$. Equivalently, we assume that

$$\frac{\gamma_{\text{DS}}}{\beta_{\text{DS}}} = \frac{\gamma_{\text{ent}}}{\beta_{\text{ent}}} = \widetilde{\gamma}.$$

According to Cauchy Inequality:

$$\left[\sum_{\tau \notin \mathcal{C}} \pi_{\text{base}}(\tau)^{1-2\widetilde{\gamma}}\right] \cdot \left[\sum_{\tau \notin \mathcal{C}} \pi_{\text{base}}(\tau)\right] \geq \left[\sum_{\tau \notin \mathcal{C}} \pi_{\text{base}}(\tau)^{1-\widetilde{\gamma}}\right]^2.$$

Thus, we have

$$\mathbb{D}_{\chi^2}(\pi_{\text{base}}\|\pi_{\text{ent}}) = -1 + \frac{\sum_{\tau \in \mathcal{C}} \pi_{\text{base}}(\tau)^{1-2\widetilde{\gamma}}}{b_{\widetilde{\gamma}}^2} \cdot C(\pi_{\text{ent}})^2 + (1 - C(\pi_{\text{ent}}))^2 \frac{\sum_{\tau \notin \mathcal{C}} \pi_{\text{base}}(\tau)^{1-2\widetilde{\gamma}}}{(B_{\widetilde{\gamma}} - b_{\widetilde{\gamma}})^2}$$

$$\geq -1 + \frac{\sum_{\tau \in \mathcal{C}} \pi_{\text{base}}(\tau)^{1-2\widetilde{\gamma}}}{b_{\widetilde{\gamma}}^2} \cdot C(\pi_{\text{ent}})^2 + (1 - C(\pi_{\text{ent}}))^2 \frac{1}{\sum_{\tau \notin \mathcal{C}} \pi_{\text{base}}(\tau)} = \mathbb{D}_{\chi^2}(\pi_{\text{base}}\|\pi_{\text{DS}}).$$

Therefore, the difference is non-negative: $\mathbb{D}_{\chi^2}(\pi_{\text{base}}\|\pi_{\text{ent}}) - \mathbb{D}_{\chi^2}(\pi_{\text{base}}\|\pi_{\text{DS}}) \geq 0$. This means that for any given correctness level $C$, our method (with $\gamma_{\text{DS}} = \gamma_{\text{ent}} \cdot \frac{\beta_{\text{DS}}}{\beta_{\text{ent}}}$) can achieve it with a lower or equal KL-divergence cost. Thus, if both methods is constrained by $\mathbb{D}_{\text{KL}}(\pi_{\text{base}}\|\pi_{\text{ent}}) \leq \kappa$ and $\mathbb{D}_{\text{KL}}(\pi_{\text{base}}\|\pi_{\text{DS}}) \leq \kappa$, our method can achieve a correctness $C(\pi_{\text{DS}}) \geq C(\pi_{\text{ent}})$. Furthermore, assuming the ratio equality $\frac{\gamma_{\text{DS}}}{\beta_{\text{DS}}} = \frac{\gamma_{\text{ent}}}{\beta_{\text{ent}}} = \widetilde{\gamma}$, the policy $\pi_{\text{DS}}$ takes the form:

$$\pi_{\text{DS}}(\tau) = \frac{1}{\mathcal{Z}} \begin{cases} [\pi_{\text{base}}(\tau)]^{1-\widetilde{\gamma}} \exp\left(\frac{1}{\beta_{\text{DS}}} r(\tau)\right) & \text{if } \tau \in \mathcal{C}, \\ [\pi_{\text{base}}(\tau)] \cdot \exp\left(\frac{1}{\beta_{\text{DS}}} r(\tau)\right) & \text{if } \tau \notin \mathcal{C}, \end{cases}$$

where $\mathcal{Z}$ is the partition function. Similarly, the entropy-regularized policy is given by:

$$\pi_{\text{ent}}^*(\tau) = \frac{[\pi_{\text{base}}(\tau)]^{1-\widetilde{\gamma}} \exp\left(\frac{1}{\beta_{\text{ent}}} r(\tau)\right)}{\sum_{\tau'} [\pi_{\text{base}}(\tau')]^{1-\widetilde{\gamma}} \exp\left(\frac{1}{\beta_{\text{ent}}} r(\tau')\right)}.$$

This yields the equivalence $\sigma(\pi_{\text{DS}}) = \sigma(\pi_{\text{ent}})$, which concludes the proof.

---

**Lemma 13** (Correctness under Forward $\chi^2$ Constraint). *When $K_\rho(\pi, \pi_{base}) = \mathbb{D}_{\chi^2}(\pi\|\pi_{base})$, for any $\gamma_{ent} \geq 0, \beta_{ent} > 0$ such that $\mathbb{D}_{\chi^2}(\pi_{ent}\|\pi_{base}) \leq \kappa$, there exist $\gamma_{DS} \geq 0$ and $\beta_{DS} > 0$ such that $\mathbb{D}_{\chi^2}(\pi_{DS}\|\pi_{base}) \leq \kappa$ and $C(\pi_{DS}) \geq C(\pi_{ent})$, and $\sigma(\pi_{DS}) \geq \sigma(\pi_{ent})$.*

---

*Proof.* The proof proceeds in three steps: we first find a functional relationship between correctness $C$ and the KL divergence for each policy, and then compare them.

**Step 1: Analyze the Entropy-Maximization Policy ($\pi_{\text{ent}}$).** The correctness of $\pi_{ent}$ is the total probability mass on correct trajectories:

$$C(\pi_{\text{ent}}) = \frac{\sum_{\tau \in \mathcal{C}} [\pi_{\text{base}}(\tau)]^{1-\frac{\gamma_{\text{ent}}}{\beta_{\text{ent}}}} \exp(1/\beta_{\text{ent}})}{\sum_{\tau \in \mathcal{T}} [\pi_{\text{base}}(\tau)]^{1-\frac{\gamma_{\text{ent}}}{\beta_{\text{ent}}}} \exp(r(\tau)/\beta_{\text{ent}})} = \frac{b_{\frac{\gamma_{\text{ent}}}{\beta_{\text{ent}}}} e^{1/\beta_{\text{ent}}}}{b_{\frac{\gamma_{\text{ent}}}{\beta_{\text{ent}}}} e^{1/\beta_{\text{ent}}} + (B_{\frac{\gamma_{\text{ent}}}{\beta_{\text{ent}}}} - b_{\frac{\gamma_{\text{ent}}}{\beta_{\text{ent}}}})}. \tag{11}$$

Solving for $e^{1/\beta_{\text{ent}}}$ yields: $e^{1/\beta_{\text{ent}}} = \frac{B_{\frac{\gamma_{\text{ent}}}{\beta_{\text{ent}}}} - b_{\frac{\gamma_{\text{ent}}}{\beta_{\text{ent}}}}}{b_{\frac{\gamma_{\text{ent}}}{\beta_{\text{ent}}}}} \left(\frac{C(\pi_{\text{ent}})}{1 - C(\pi_{\text{ent}})}\right).$

The reverse $\chi^2$ divergence $\mathbb{D}_{\chi^2}(\pi_{\text{ent}}\|\pi_{\text{base}})$ can be expressed as a function of $C(\pi_{\text{ent}})$. Following the derivation previously, we arrive at:

$$\mathbb{D}_{\chi^2}(\pi_{\text{ent}}\|\pi_{\text{base}}) = -1 + \left[\sum_\tau \pi_{\text{base}}(\tau)^{1-\frac{\gamma_{\text{ent}}}{\beta_{\text{ent}}}} \exp\left(\frac{r(\tau)}{\beta_{\text{ent}}}\right)\right] \cdot \left[\sum_\tau \pi_{\text{base}}(\tau)^{1+\frac{\gamma_{\text{ent}}}{\beta_{\text{ent}}}} \exp\left(-\frac{r(\tau)}{\beta_{\text{ent}}}\right)\right]$$

$$= \frac{1}{C(\pi_{\text{ent}})} b_{\frac{\gamma_{\text{ent}}}{\beta_{\text{ent}}}} \sum_{\tau \in \mathcal{C}} \pi_{\text{base}}(\tau)^{1+\frac{\gamma_{\text{ent}}}{\beta_{\text{ent}}}} + \frac{1}{1 - C(\pi_{\text{ent}})} (B_{\frac{\gamma_{\text{ent}}}{\beta_{\text{ent}}}} - b_{\frac{\gamma_{\text{ent}}}{\beta_{\text{ent}}}}) \sum_{\tau \notin \mathcal{C}} \pi_{\text{base}}(\tau)^{1-\frac{\gamma_{\text{ent}}}{\beta_{\text{ent}}}}$$

**Step 2: Analyze Differential Policy ($\pi_{\text{DS}}$).** Similarly, the correctness for our policy is given by:

$$C(\pi_{\text{DS}}) = \frac{b_{\frac{\gamma_{\text{DS}}}{\beta_{\text{DS}}}} e^{1/\beta_{\text{DS}}}}{b_{\frac{\gamma_{\text{DS}}}{\beta_{\text{DS}}}} e^{1/\beta_{\text{DS}}} + (1 - p_c)}. \tag{12}$$

The corresponding reverse $\chi^2$ divergence as a function of $C(\pi_{\text{DS}})$ is:

$$\mathbb{D}_{\chi^2}(\pi_{\text{ent}} \| \pi_{\text{base}}) = \frac{1}{C(\pi_{\text{ent}})} b_{\frac{\gamma_{\text{DS}}}{\beta_{\text{DS}}}} \sum_{\tau \in \mathcal{C}} \pi_{\text{base}}(\tau)^{1+\frac{\gamma_{\text{DS}}}{\beta_{\text{DS}}}} + \frac{1}{(1 - C(\pi_{\text{ent}}))} \left((1 - p_c)\right) \sum_{\tau \notin \mathcal{C}} \pi_{\text{base}}(\tau)^{1+\frac{\gamma_n}{\beta_{\text{DS}}}}$$

**Step 3: Compare the Policies.** Our goal is to show that for any $C(\pi_{\text{ent}})$, we can choose parameters for our method to achieve $C(\pi_{\text{DS}}) \geq C(\pi_{\text{ent}})$ with a smaller or equal $\chi^2$ divergence. Let's choose $\gamma_{\text{DS}} = \frac{\beta_{\text{DS}}}{\beta_{\text{ent}}} \gamma_{\text{ent}}$, and set $C(\pi_{\text{DS}}) = C(\pi_{\text{ent}}) = C$. Equivalently, we assume that

$$\frac{\gamma_{\text{DS}}}{\beta_{\text{DS}}} = \frac{\gamma_{\text{ent}}}{\beta_{\text{ent}}} = \widetilde{\gamma}.$$

. According to Cauchy Inequality:

$$\left[\sum_{\tau \notin \mathcal{C}} \pi_{\text{base}}(\tau)^{1+\widetilde{\gamma}}\right] \cdot \left[\sum_{\tau \notin \mathcal{C}} \pi_{\text{base}}(\tau)^{1-\widetilde{\gamma}}\right] \geq \left[\sum_{\tau \notin \mathcal{C}} \pi_{\text{base}}(\tau)\right]^2.$$

Thus, we have

$$\mathbb{D}_{\chi^2}(\pi_{\text{ent}} \| \pi_{\text{base}}) = \frac{1}{C(\pi_{\text{ent}})} b_{\widetilde{\gamma}} \sum_{\tau \in \mathcal{C}} \pi_{\text{base}}(\tau)^{1+\widetilde{\gamma}} + \frac{1}{1 - C(\pi_{\text{ent}})} (B_{\widetilde{\gamma}} - b_{\widetilde{\gamma}}) \sum_{\tau \notin \mathcal{C}} \pi_{\text{base}}(\tau)^{1-\widetilde{\gamma}}$$

$$\geq \frac{1}{C(\pi_{\text{ent}})} b_{\widetilde{\gamma}} \sum_{\tau \in \mathcal{C}} \pi_{\text{base}}(\tau)^{1+\widetilde{\gamma}} + \frac{1}{1 - C(\pi_{\text{ent}})} \left(\sum_{\tau \notin \mathcal{C}} \pi_{\text{base}}(\tau)\right)^2 = \mathbb{D}_{\chi^2}(\pi_{\text{ent}} \| \pi_{\text{base}})$$

Therefore, the difference is non-negative: $\mathbb{D}_{\chi^2}(\pi_{\text{ent}} \| \pi_{\text{base}}) - \mathbb{D}_{\chi^2}(\pi_{\text{DS}} \| \pi_{\text{base}}) \geq 0$. This means that for any given correctness level $C$, our method (with $\gamma_{\text{DS}} = \gamma_{\text{ent}} \cdot \frac{\beta_{\text{DS}}}{\beta_{\text{ent}}}$) can achieve it with a lower or equal KL-divergence cost. Thus, if the entropy method is constrained by $\mathbb{D}_{\chi^2} \leq \kappa$, our method can achieve a correctness $C(\pi_{\text{DS}}) \geq C(\pi_{\text{ent}})$ while also satisfying the constraint. Furthermore, assuming the ratio equality $\frac{\gamma_{\text{DS}}}{\beta_{\text{DS}}} = \frac{\gamma_{\text{ent}}}{\beta_{\text{ent}}} = \widetilde{\gamma}$, the policy $\pi_{\text{DS}}$ takes the form:

$$\pi_{\text{DS}}(\tau) = \frac{1}{\mathcal{Z}} \begin{cases} [\pi_{\text{base}}(\tau)]^{1-\widetilde{\gamma}} \exp\left(\frac{1}{\beta_{\text{DS}}} r(\tau)\right) & \text{if } \tau \in \mathcal{C}, \\ [\pi_{\text{base}}(\tau)] \cdot \exp\left(\frac{1}{\beta_{\text{DS}}} r(\tau)\right) & \text{if } \tau \notin \mathcal{C}, \end{cases}$$

where $\mathcal{Z}$ is the partition function. Similarly, the entropy-regularized policy is given by:

$$\pi_{\text{ent}}^*(\tau) = \frac{[\pi_{\text{base}}(\tau)]^{1-\widetilde{\gamma}} \exp\left(\frac{1}{\beta_{\text{ent}}} r(\tau)\right)}{\sum_{\tau'} [\pi_{\text{base}}(\tau')]^{1-\widetilde{\gamma}} \exp\left(\frac{1}{\beta_{\text{ent}}} r(\tau')\right)}.$$

This yields the equivalence $\sigma(\pi_{\text{DS}}) = \sigma(\pi_{\text{ent}})$, which concludes the proof. $\qquad\square$

**Theorem 14.** *Assume the reward mechanism has access to all correct trajectories. For any parameters $\gamma_{ent} \geq 0$ and $\beta_{ent} > 0$ used in the entropy-regularized policy $\pi_{ent}$ that satisfy a proximity constraint $K_\rho(\pi_{ent}, \pi_{base}) \leq \kappa$, there exist parameters $\gamma_{DS} \geq 0$ and $\beta_{n,p} > 0$ for our proposed policy $\pi_{DS}$ such that it also satisfies $K_\rho(\pi_{DS}, \pi_{base}) \leq \kappa$, and the following inequalities hold:*

$$C(\pi_{DS}) \geq C(\pi_{ent}) \quad \text{and} \quad \sigma(\pi_{DS}) \leq \sigma(\pi_{ent}).$$

*This result holds for divergence measures $K_\rho(\pi, \pi_{base})$ including $\mathbb{D}_{KL}(\pi \| \pi_{base})$, $\mathbb{D}_{KL}(\pi_{base} \| \pi)$, $\mathbb{D}_{\chi^2}(\pi \| \pi_{base})$, and $\mathbb{D}_{\chi^2}(\pi_{base} \| \pi)$.*

*Proof.* According to Lemma 10, Lemma 11, Lemma 12, and Lemma 13, we obtain that Theorem 14 holds for $K_\rho(\pi, \pi_{\text{base}}) = \mathbb{D}_{\text{KL}}(\pi \| \pi_{\text{base}}), \mathbb{D}_{\text{KL}}(\pi_{\text{base}} \| \pi), \mathbb{D}_{\chi^2}(\pi \| \pi_{\text{base}}), \mathbb{D}_{\chi^2}(\pi_{\text{base}} \| \pi)$. Thus, we finish the proof of the theorem. $\qquad\square$

**B.3. Equivalence of Theoretical and Practical Reward Modifications**

In this section, we clarify the relationship between our theoretical reward modification and its practical implementation. Specifically, we demonstrate that subtracting a $\log \pi$ term from the reward is equivalent to subtracting a $\log \pi_{\text{base}}$ term, under a re-parameterization of the optimization objective.

Consider the following theoretical reward modification, which uses the policy's own probability $\pi$:

$$r_{\text{DS}}^{\pi}(\tau) = \begin{cases} r(\tau) - \gamma_p \cdot \log(\pi(\tau)) & \text{if } r(\tau) > 0 \quad \text{(correct trajectories)} \\ r(\tau) + \gamma_n \cdot \log(\pi(\tau)) & \text{if } r(\tau) \leq 0 \quad \text{(incorrect trajectories).} \end{cases} \tag{13}$$

We first define the theoretical optimization problem for parameters $\beta, \gamma_n, \gamma_p$:

$$\pi_{\text{DS}} = \arg \max_{\pi} \mathbb{E}_{\tau \sim \pi} \left[ r_{\text{DS}}^{\pi}(\tau) \right] - \beta \cdot \mathbb{D}_{\text{KL}}(\pi || \pi_{\text{base}}). \tag{14}$$

Now, consider an alternative formulation where the reward is modified using the base policy $\pi_{\text{base}}$:

$$r_{\text{DS}}^{\pi_{\text{base}}}(\tau) = \begin{cases} r(\tau) - \widetilde{\gamma}_p \cdot \log(\pi_{\text{base}}(\tau)) & \text{if } r(\tau) > 0 \quad \text{(correct trajectories)} \\ r(\tau) + \widetilde{\gamma}_n \cdot \log(\pi_{\text{base}}(\tau)) & \text{if } r(\tau) \leq 0 \quad \text{(incorrect trajectories).} \end{cases} \tag{15}$$

We show that the solution $\pi_{\text{DS}}$ to the original problem Equation (14) is **also** the solution to the following practical objective, which uses $r_{\text{DS}}^{\pi_{\text{base}}}$:

$$\pi_{\text{DS}} = \arg \max_{\pi} \mathbb{E}_{\tau \sim \pi} \left[ r_{\text{DS}}^{\pi_{\text{base}}}(\tau) \right] - \widetilde{\beta} \cdot \mathbb{D}_{\text{KL}}(\pi || \pi_{\text{base}}). \tag{16}$$

This equivalence holds when the new parameters $\widetilde{\beta}, \widetilde{\gamma}_p$, and $\widetilde{\gamma}_n$ are set as follows:

$$\widetilde{\beta} = \beta + \gamma_p, \quad \widetilde{\gamma}_p = \gamma_p, \quad \widetilde{\gamma}_n = \frac{\gamma_n(\beta + \gamma_p)}{\beta + \gamma_n}. \tag{17}$$

Therefore, the theoretical analysis from Theorem 4 still holds when $\log \pi$ is substituted for $\log \pi_{\text{base}}$. Furthermore, the policy selection mechanism in Eq. 4 is equivalent to directly maximizing entropy via an added regularization term. In practice, we find that using $\log \pi_{\theta_{\text{old}}}$ (i.e., the log-probability of a previous policy iteration) in the advantage function modification yields empirically better performance than using $\log \pi_{\text{base}}$ (the log-probability of the base policy).

# C. Experimental Details

In this section, we provide additional details for the experiments in Section 3.

## C.1. Countdown Experiment

### C.1.1. DATA

We use the dataset released by Pan et al. (2025), which contains 327,680 training samples and 1,024 test samples.[1] An example training prompt is shown below.

> **Countdown Task Example**
>
> A conversation between User and Assistant. The user asks a question, and the Assistant solves it. The assistant first thinks about the reasoning process in the mind and then provides the user with the answer.
> **User:** Using the numbers [5, 94, 9, 44], create an equation that equals 93. You can use basic arithmetic operations (+, -, *, /) and each number can only be used once. Show your work in `<think> </think>` tags. And return the final answer in `<answer> </answer>` tags, for example

---

[1] https://huggingface.co/datasets/Jiayi-Pan/Countdown-Tasks-3to4

```
<answer> (1 + 2) / 3 </answer>.
Assistant: Let me solve this step by step.
<think>
```

Our implementation builds on the official repository of Pan et al. (2025)[2] and a fork adapted for A100 training.[3]

### C.1.2. TRAINING

We train with a global batch size of $128$, with $5$ rollouts per prompt, and use a mini-batch size of $64$. The learning rate is $1 \times 10^{-6}$, and the KL penalty coefficient is set to $\beta_{\mathrm{KL}} = 1 \times 10^{-3}$. The reward is $1$ for correct responses, $0.1$ for incorrect yet properly formatted responses, and $0$ for all others. The maximum response length is $8{,}192$ tokens. We perform RL fine-tuning of the `Qwen2.5-3B-Instruct` model (Hui et al., 2024) for $320$ steps on $2$ A100 GPUs.

| Configuration for Qwen3-1.7B | | | |
|---|---|---|---|
| **Parameter** | **Value** | **Parameter** | **Value** |
| Pretrained model | Qwen3-1.7B | Training set | DAPO14k |
| Prompts per batch | 32 | Generations/prompt | 8 |
| Gradient update/step | 1 | Max prompt length | 1024 |
| Max response length | 4096 | Learning rate | $5 \times 10^{-7}$ |
| Clip ratio low | 0.2 | Clip ratio high | 0.25 |
| Training Steps | 300 | $\beta$ | 0.0 |
| Entropy coefficient | 0.0 | $\gamma_p$ | 0.02 |
| $\gamma_n$ | 0.02 | Remove padding | Enabled |
| Rollout engine | vllm | Rollout temperature | 1.0 |
| Validation temperature | 0.7 | Device | 4 x Nvidia-H100 |

| Configuration for Qwen2.5-Math-1.5B | | | |
|---|---|---|---|
| **Parameter** | **Value** | **Parameter** | **Value** |
| Pretrained Model | Qwen2.5-Math-1.5B | Training Set | DAPO14k |
| Prompts per batch | 32 | Generations/prompt | 8 |
| Gradient update/step | 1 | Max prompt length | 1024 |
| Max response length | 2048 | Learning rate | $1 \times 10^{-6}$ |
| Clip ratio low | 0.2 | Clip ratio high | 0.2 |
| Training Steps | 1000 | $\beta$ | 0.0 |
| Entropy coefficient | 0.0 | $\gamma_p$ | 0.05 |
| $\gamma_n$ | 0.05 | Remove padding | Enabled |
| Rollout engine | vllm | Rollout temperature | 1.0 |
| Validation temperature | 0.7 | Device | 4 x Nvidia-L6000 |

For Qwen2.5-Math-7B model, we trained for three random seeds. During evaluation, we first generated $128$ rollouts for each question, then estimated Pass@1 to Pass@64 using the unbiased estimator of each metric respectively, following Walder & Karkhanis (2026).

---

[2] https://github.com/Jiayi-Pan/TinyZero
[3] https://github.com/JerryWu-code/TinyZero

| Configuration for Qwen2.5-Math-7B | | | |
|---|---|---|---|
| **Parameter** | **Value** | **Parameter** | **Value** |
| Pretrained Model | Qwen2.5-Math-7B | Training Set | DAPO14k |
| Prompts per batch | 32 | Generations/prompt | 8 |
| Gradient update/step | 1 | Max prompt length | 1024 |
| Max response length | 2048 | Learning rate | $1 \times 10^{-6}$ |
| Clip ratio low | 0.2 | Clip ratio high | 0.2 |
| Training Steps | 500 | $\beta$ | 0.0 |
| Entropy coefficient | 0.0 | $\gamma_p$ | 0.05 |
| $\gamma_n$ | 0.05 | Remove padding | Enabled |
| Rollout engine | `vllm` | Rollout temperature | 1.0 |
| Validation temperature | 0.7 | Device | 4 x Nvidia-A100 |

| Configuration for Ministral-8B-Instruct | | | |
|---|---|---|---|
| **Parameter** | **Value** | **Parameter** | **Value** |
| Pretrained model | Ministral-8B-Instruct | Training set | DAPO14k + MATH12k |
| Prompts per batch | 128 | Generations/prompt | 8 |
| Gradient update/step | 4 | Max prompt length | 1024 |
| Max response length | 2048 | Learning rate | $3 \times 10^{-7}$ |
| Clip ratio low | 0.2 | Clip ratio high | 0.22 |
| Training steps | 300 | $\beta$ | 0.0 |
| Entropy coefficient | 0.0 | $\gamma_p$ | 0.02 |
| $\gamma_n$ | 0.02 | Remove padding | Enabled |
| Rollout engine | `vllm` | Rollout temperature | 1.0 |
| Validation temperature | 0.7 | Device | 4 x Nvidia-A100 |

## C.2. Baseline Implementation

**GR-PKPO.** We attempted to train the model using the Pass@$k$ metric directly as the reward signal for $k \in \{2, 3, 4\}$. However, this approach proved unstable across all configurations. The training process quickly collapsed, causing the model to generate degenerate outputs and yielding performance substantially worse than the baseline. Consequently, these results are omitted from our main comparisons. We hypothesize this instability may be attributed to the limited number of rollouts (5) used during training.

**Unlikeliness Reward Method.** We compare against the rank-based penalty proposed by He et al. (2025a), which down-weights the reward for high-probability solutions to encourage diversity. For a set of rollouts $\{y_i\}_{i=1}^{G}$, the modified reward is:

$$r_{\text{unlikely}}(y_i) = r(y_i) \left(1 - \beta_{\text{rank}} \frac{G - \text{rank}(y_i)}{G}\right),$$

where $\text{rank}(y_i)$ is the rank of the solution $y_i$ based on its probability under the policy $\pi_{\text{old}}(\cdot \mid x)$. We performed a hyperparameter sweep for $\beta_{\text{rank}} \in \{0.05, 0.1, 0.15, 0.2, 0.25, 0.3\}$ and found that $\beta_{\text{rank}} = 0.05$ yielded the best Pass@64 performance. For values of $\beta_{\text{rank}} \geq 0.15$, the training became unstable and collapsed.

**Entropy-Based Methods.** For the entropy-based baselines, we augment GRPO with either an entropy bonus or penalty. We swept the coefficients $\eta_+$ and $\eta_-$ over the set $\{0.005, 0.01, 0.03, 0.05\}$. In Figure 5, we report the results for the hyperparameter value that achieved the highest Pass@64 score. A comprehensive summary of all experimental runs is available in Appendix D.

## C.3. Math Reasoning Experiment

We first filter out 14,042 questions written in English from the DAPO dataset (Yu et al., 2026) (which we refer to as DAPO14k). We then conduct reinforcement fine-tuning on four models: `Qwen2.5-Math-1.5B` (Hui et al., 2024), `Qwen2.5-Math-7B` (Hui et al., 2024), and `Qwen3-1.7B` (Yang et al., 2025) are trained on DAPO14k; `Ministral-8B-Instruct` (Jiang et al., 2023) is trained on DAPO14k augmented with 11,998 questions from the MATH12k dataset (Hendrycks et al., 2021). The detailed per-model training setups are shown in the tables above and below; response examples are shown in the boxes below.

---

**Mistral Instruction Response Example**

**[INST]** Convert the point $(0, 3)$ in rectangular coordinates to polar coordinates. Enter your answer in the form $(r, \theta)$, where $r > 0$ and $0 \leq \theta < 2\pi$. Let's think step by step and output the final answer within `\boxed{}`. **[/INST]**

1. **Calculate the radius $r$:**

$$r = \sqrt{0^2 + 3^2} = \sqrt{0 + 9} = \sqrt{9} = 3.$$

2. **Determine the angle $\theta$:**

   - The point $(0, 3)$ is on the positive $y$-axis.

   - The angle $\theta$ with the positive $x$-axis is given by:

$$\theta = \tan^{-1}\left(\frac{3}{0}\right).$$

   - Since the point is on the positive $y$-axis, the angle is $\frac{\pi}{2}$.

3. **Combine the results to get the polar coordinates:**

$$\boxed{(3, \frac{\pi}{2})}$$

---

**Qwen2.5-Math-1.5B**

`<|im_start|>system`
Please reason step by step, and put your final answer within `\boxed{}`. `<|im_end|>`
`<|im_start|>user`
Cities $A$ and $B$ are $45$ miles apart. Alicia lives in $A$ and Beth lives in $B$. Alicia bikes towards $B$ at 18 miles per hour. Leaving at the same time, Beth bikes toward $A$ at 12 miles per hour. How many miles from City $A$ will they be when they meet? Let's think step by step and output the final answer within `\boxed{}`. `<|im_end|>`
`<|im_start|>assistant`
To determine where Alicia and Beth will meet, we need to find out how long it will take for them to meet and then use that time to calculate the distance Alicia has traveled.

1. First, let's calculate their combined speed. Since Alicia bikes at 18 miles per hour and Beth bikes at 12 miles per hour, their combined speed is $18 + 12 = 30$ miles per hour.

2. Since the distance between the two cities is 45 miles, the time it takes for them to meet is the total distance divided by their combined speed. So, the time $t$ it takes for them to meet is

$$t = \frac{45}{30} = 1.5 \text{ hours.}$$

3. Now, we need to find out how far Alicia has traveled in 1.5 hours. Since Alicia bikes at 18 miles

per hour, the distance she travels in 1.5 hours is

$$18 \times 1.5 = 27 \text{ miles}.$$

So, Alicia and Beth will meet 27 miles from City $A$. The final answer is

$$\boxed{27}$$

**<|im_end|>**

| **Configuration for Countdown Task** | | | |
|---|---|---|---|
| **Parameter** | **Value** | **Parameter** | **Value** |
| Pretrained model | Qwen2.5-3B-Instruct | Batch size | 128 |
| Generations/prompt | 5 | Mini-batch size | 64 |
| Max prompt length | 2,048 | Max response length | 8,192 |
| Learning rate | $1 \times 10^{-6}$ | Training steps | 320 |
| Entropy coefficient | 0.001 | Clip ratio | 0.2 |
| $\gamma_p$ | 0.03 | $\gamma_n$ | 0.01 |
| Rollout engine | `vllm` | Rollout temperature | 1 |
| Validation temperature | 1 | Validation top-k | 50 |
| Validation top-p | 0.7 | Device | 2 x A100 |

## C.4. Experimental Details for Section 3.7

In this section, we present experimental details to substantiate the claim made in Section 3.7: namely, that the Solution Multiplicity $\text{Solution Multiplicity}(\mathcal{X})$, the average number of unique solutions per problem instance, serves as an appropriate metric for evaluating the benefits of encouraging policy entropy in a task. The tasks we consider are as follows:

- **Math Reasoning**: As described in Appendix C.3.

- **Countdown**: As described in Appendix C.1.

- **Countdown-3**: A variant of Countdown where we filter the dataset to retain only samples with three given numbers.

- **Knights and Knaves**: A classic logic puzzle in which the objective is to identify truth-tellers (knights) and liars (knaves) from their statements. (Stojanovski et al., 2026). Below is an example question and correct response.

---

**Knights-and-Knaves**

**Question:**
A very special island is inhabited only by sages and fools. Sages always tell the truth, and fools always lie. You meet 6 inhabitants: Owen, Liam, Evelyn, William, Sofia, and Emily. As Owen put it, "William is a sage". Liam was heard saying, "William is a fool". Evelyn noted, "Sofia is a fool and Evelyn is a sage". "if Owen is a sage then Sofia is a sage," William mentioned. "if Liam is a sage then Evelyn is a sage," Sofia claimed. Emily told you that if Emily is a sage then Liam is a fool. So who is a sage and who is a fool? (Format your answer like: "Owen is a sage/fool, Liam is a sage/fool, Evelyn is a sage/fool, William is a sage/fool, Sofia is a sage/fool, and Emily is a sage/fool") Let's think step by step and output the final answer (should be a sentence) within \boxed{}.
**Response:**

- We need to consider two cases for Owen:
    - *Case 1: Owen is a sage.* If Owen is a sage, then his statement is true. Therefore, William must be a sage.
    - *Case 2: Owen is a fool.* If Owen is a fool, then his statement is false. Therefore, William must be a fool.

- **Liam's statement**: "William is a fool."
    - *Case 1: Liam is a sage.* If Liam is a sage, then his statement is true. Therefore, William must be a fool.

---

> – *Case 2: Liam is a fool.* If Liam is a fool, then his statement is false. Therefore, William must be a sage.
>
> - **William's statement**: "If Owen is a sage then Sofia is a sage."
>   Since Sofia is a sage, William's statement is true. Therefore, William must be a sage.
>
> - **Owen's statement revisited**: "William is a sage."
>   Since William is a sage, Owen's statement is true. Therefore, Owen must be a sage.
>
> - **Liam's statement revisited**: "William is a fool."
>   Since William is a sage, Liam's statement is false. Therefore, Liam must be a fool.
>
> - **Emily's statement revisited**: "If Emily is a sage then Liam is a fool."
>   Since Emily is a sage (from earlier), according to her statement, Liam must be a fool. This confirms our previous determination.
>
> - **Owen's initial statement**: "William is a sage."
>   Since Owen is a sage and William is a sage, Owen's statement is true. This is consistent.
>
> **Therefore, summarizing:**
>
> - Owen is a sage.
>
> - Liam is a fool.
>
> - Evelyn's statement was a contradiction, so Evelyn must be a fool.
>
> - William is a sage.
>
> - Sofia is a sage.
>
> - Emily's statement was consistent, so Emily must be a sage.
>
> > $Owen$ is a sage, $Liam$ is a fool, $Evelyn$ is a fool, $William$ is a sage, $Sofia$ is a sage, and $Emily$ is a sage

To estimate Solution Multiplicity, we query the GPT-5-Think model on 200 randomly selected samples with the following prompt:

> **Prompt for Querying Solution Multiplicity**
>
> **User:** You are an expert mathematics educator and problem solver. Analyze the given mathematical problem and determine how many different solution approaches exist for it.
> Please provide a comprehensive analysis that: 1. Identifies all distinct solution methods/approaches 2. Briefly explains what each approach involves 3. Counts the total number of different approaches
> Mathematical Problem: problem
> Please first explain what different solution approaches exist for this problem, then provide your final answer in the format: ¡ways¿ [number] ¡/ways¿
> For example, if a problem has exactly 2 different solution methods, your response should end with: ¡ways¿ 2 ¡/ways¿

*Table 1.* Illustration of Experimental Result

| Task | Knights | Math | Count-3 | Count |
|------|---------|------|---------|-------|
| Solution Multiplicity | 1.5 | 3.7 | 6.5 | 15.7 |
| Pass@8 of GRPO | 47.1 | 78.6 | 97.7 | 73.4 |
| Pass@8 (GRPO + Ent) | 38.1 | 72.6 | 98.7 | 76.8 |
| **Entropy Effect** | **-9.0%** | **-6.0%** | **+1.0%** | **+3.4%** |

**Training details.** The training setups of Math reasoning and Countdown task are identical to the main experiments as described in Appendix C.1 and C.3. For Countdown-3, we train the model for 160 steps. For GRPO with entropy bonus, we use a bonus coefficient of $\eta_+ = 0.05$. Other configurations are identical to those of the main experiment. For Knights-and-Knaves, we RL fine-tune the Qwen2.5-7B-Instruct (Hui et al., 2024) model with LoRA adaptation (rank 256) (Hu et al., 2022) for 100 steps. We use a learning rate of $4 \times 10^{-5}$ and a batch size of 32, with 8 rollouts per sample.

**Result Analysis.** The experimental results, presented in Table 1, reveal a direct correlation between Solution Multiplicity and the efficacy of entropy regularization. Specifically, as a task's Solution Multiplicity increases, so does the performance gain (Pass@8) of an entropy bonus over vanilla GRPO. This provides strong empirical support for our hypothesis: for tasks with a larger solution space, the benefits of enhanced diversity outweigh the potential trade-offs in single-solution correctness. These findings thus validate Solution Multiplicity as a practical metric for guiding the decision of whether to increase or decrease entropy for a given task.

# D. Additional Experimental Results for Countdown

In this section, we provide additional results for the Countdown task.

### D.1. Additional Experiments for Entropy coefficient

To provide a more comprehensive comparison, we analyze the performance of the entropy-based baselines across their full hyperparameter sweep. We compare DS-GRPO against GRPO with varying entropy bonus ($\eta_+$) and penalty ($\eta_-$) coefficients, with the results illustrated in Figure 6 (Top). The figure clearly demonstrates that DS-GRPO consistently outperforms the global entropy control methods across their entire range of tested hyperparameters for all values of $K$.

### D.2. Effects of KL Coefficient and Other Factors

Figure 6 (Bottom Left) reports results with varying sampling temperatures for both DS-GRPO and GRPO. Under the same temperature, DS-GRPO achieves consistently higher Pass@$K$.

Figure 6 (Bottom Right) presents results from varying the unlikeliness reward coefficient $\beta_{\text{rank}} \in \{0.05, 0.1, 0.15, 0.2, 0.25, 0.3\}$ (He et al., 2025a). For $\beta_{\text{rank}} \geq 0.15$, training collapses and accuracy drops to 0, so we omit those results.

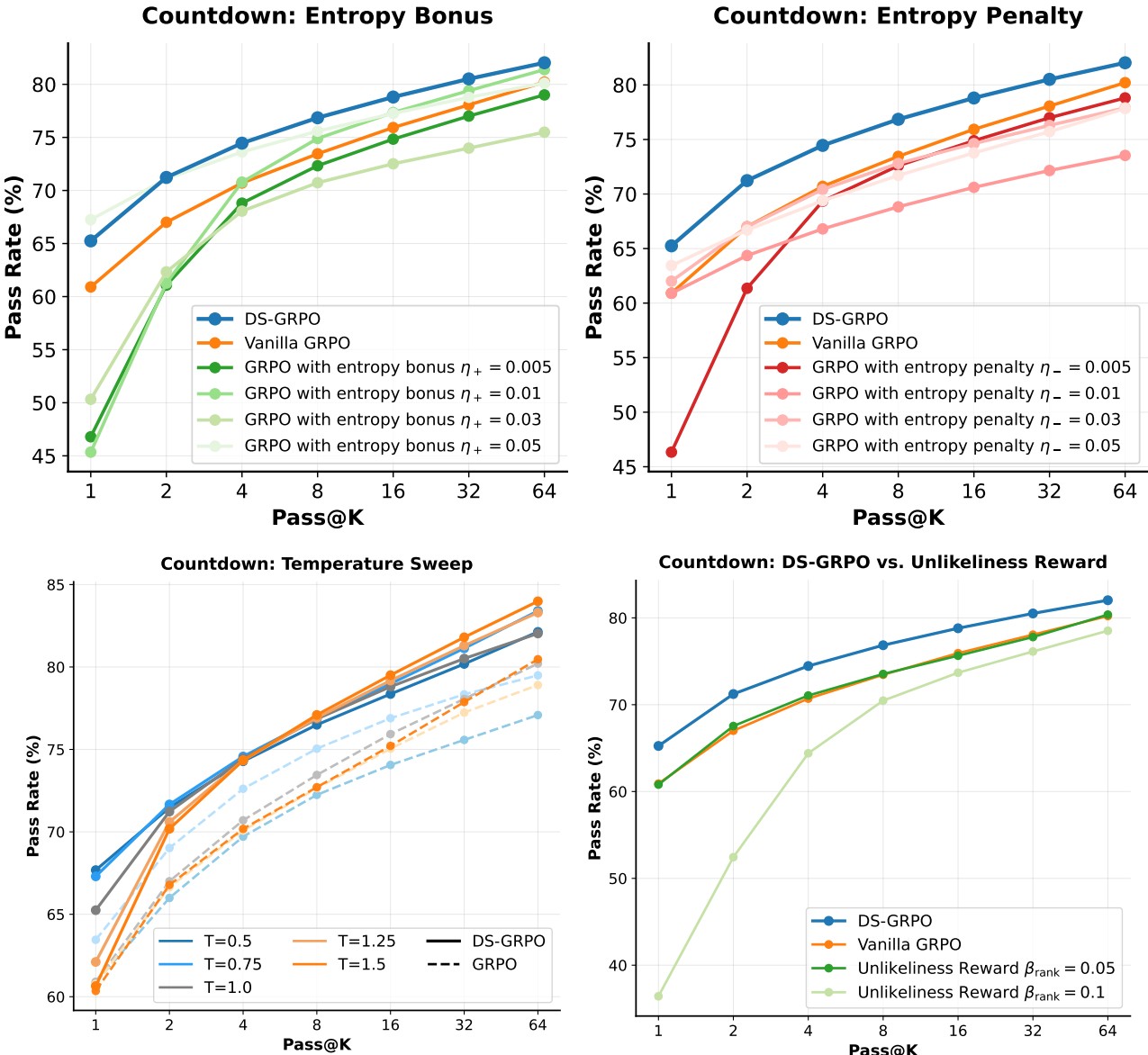

*Figure 6.* Additional results on the Countdown task comparing DS-GRPO with various baselines. Top: Pass@$K$ performance of DS-GRPO and GRPO under entropy bonus and entropy penalty. Bottom Left: Pass@$K$ performance of DS-GRPO and GRPO across different sampling temperatures. Bottom Right: Pass@$K$ performance of DS-GRPO and the Unlikeliness Reward method with varying coefficients.

# E. Additional Experimental Results for Math Reasoning Experiment

## E.1. Additional Experimental Results on DS-GRPO vs GRPO

**Experimental Results.** Figure 7, which contains the full results for Section 3.4, compares our proposed DS-GRPO against the vanilla GRPO baseline across four different base models and five mathematical reasoning benchmarks. The results demonstrate that DS-GRPO outperforms vanilla GRPO across most tested (model, dataset) pairs.

## E.2. Additional Ablation Study for DS-GRPO

**Ablation Study Implementation.** To isolate the contribution of each component in our reward modification strategy, we conduct an ablation study. We compare the full DS-GRPO algorithm against two specialized variants: *DS-GRPO-Positive*,

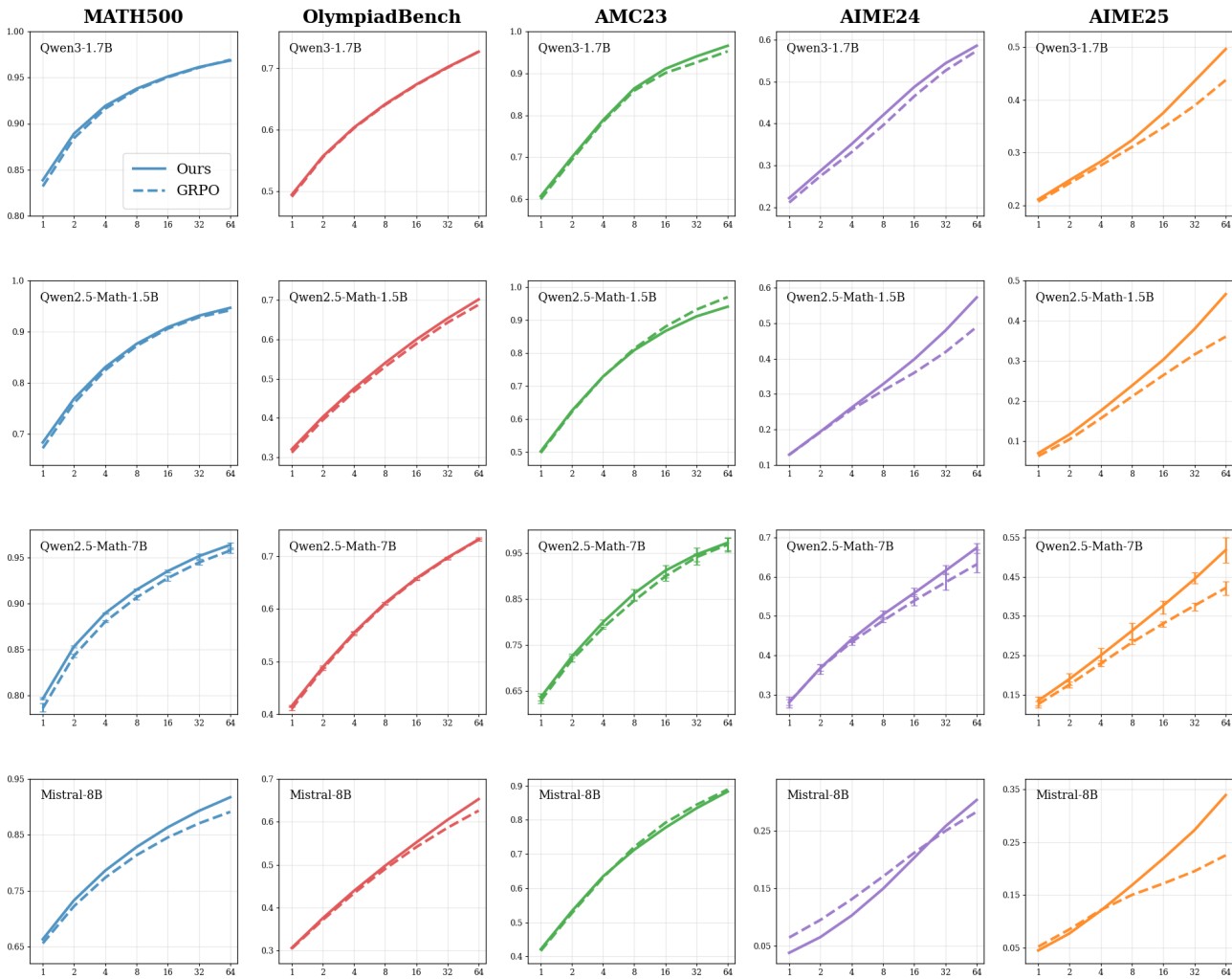

*Figure 7.* Pass@K performance after reward modification, compared with vanilla GRPO. X-axis denotes K and y-axis denotes pass rates. Trained on the DAPO(Yu et al., 2026) and the MATH(Hendrycks et al., 2021) Dataset.

which only modifies the advantage for correct trajectories, and *DS-GRPO-Negative*, which only modifies the advantage for incorrect trajectories.

Their respective advantage modifications are defined as follows:

$$A_i^{\text{DS+}} = A_i - \gamma_p \, \log \pi_{\theta_{\text{old}}}(y_i \mid x), \quad \text{if } r_i = 1,$$
$$A_i^{\text{DS-}} = A_i + \gamma_n \, \log \pi_{\theta_{\text{old}}}(y_i \mid x), \quad \text{if } r_i \neq 1.$$

The DS-GRPO-Positive variant applies only the modification to correct trajectories ($A_i^{\text{DS+}}$), leaving the advantage for incorrect trajectories as the standard $A_i$. Conversely, the DS-GRPO-Negative variant applies only the modification to incorrect trajectories ($A_i^{\text{DS-}}$), leaving the advantage for correct trajectories unchanged. **Result Analysis.** We present the results of our ablation study in Figure 8. The key findings are as follows:

- *DS-GRPO-Positive vs. Vanilla GRPO.* As shown in Figure 8, DS-GRPO-Positive outperforms vanilla GRPO, particularly for larger values of $K$. This demonstrates that modifying the reward for correct trajectories successfully mitigates the sharpening effect, providing empirical support for our intuition in Section 2.4 that penalizing high-probability correct solutions enhances diversity.

- *DS-GRPO-Negative vs. Vanilla GRPO.* The figure also shows that DS-GRPO-Negative consistently outperforms vanilla GRPO across all values of $K$. This indicates that modifying the reward for incorrect trajectories is effective at improving

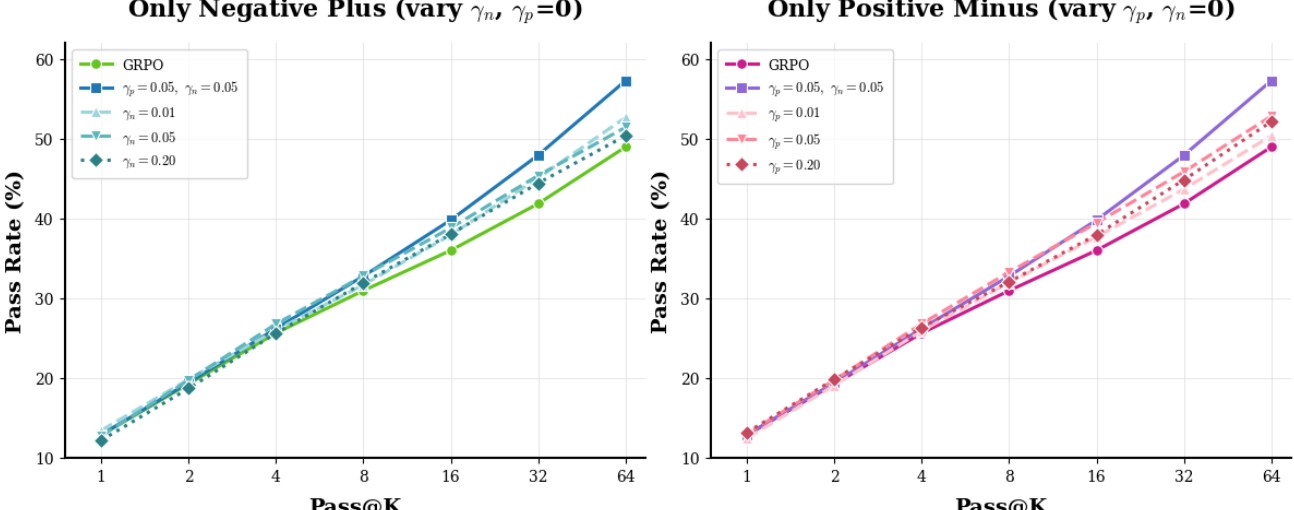

*Figure 8.* Comparison of DS-GRPO with single-side ablations on AIME24 (Qwen2.5-Math-1.5B). Left: *DS-GRPO-Negative* only ($\gamma_p = 0$, vary $\gamma_n$). Right: *DS-GRPO-Positive* only ($\gamma_n = 0$, vary $\gamma_p$). The combined DS-GRPO (blue/purple solid) dominates each single-side variant at large $K$.

the model's overall correctness.

- *DS-GRPO vs. Its Components.* The full DS-GRPO algorithm demonstrates superior performance over both of its individual components (DS-GRPO-Positive and DS-GRPO-Negative) for all $K$. This highlights a clear synergy: the "Positive" component drives diversity, while the "Negative" component enhances correctness. Their combination in DS-GRPO achieves the best balance, validating our complete reward modification strategy as outlined in Section 2.4.

### E.3. Additional Experiment on Comparing DS-GRPO with CISPO

We compare DS-GRPO with CISPO (Chen et al., 2025a), as illustrated in Figure 9. The results demonstrate that DS-GRPO achieves a higher Pass@K than CISPO at most $K$ across all datasets, with the largest gains at large $K$. All experiments were conducted using the Qwen2.5-Math-1.5B model.

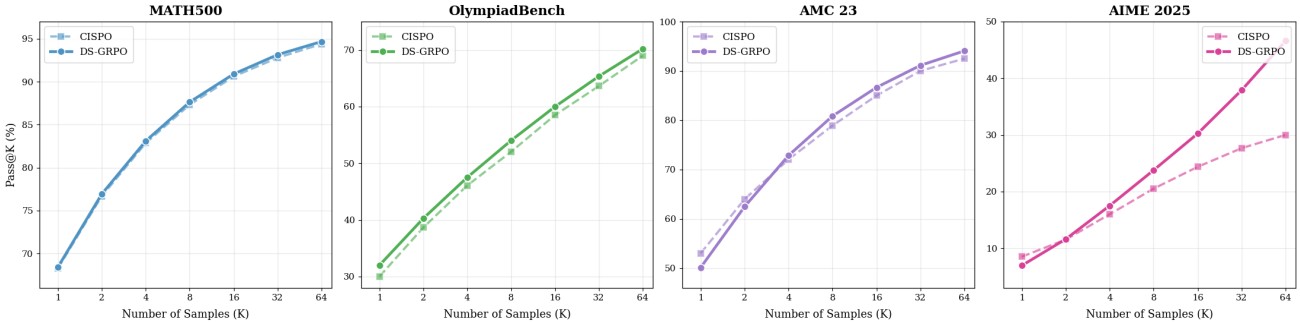

*Figure 9.* Comparison on CISPO with DS-GRPO.

# F. Supplementary Experiments and Analysis

### F.1. Diversity Changes of DS-GRPO Comparing to Baseline Model

To provide stronger evidence that DS-GRPO truly mitigates entropy collapse, follow the convention of (Hochlehnert et al., 2025). We plot the Pass@K difference between DS-GRPO with base model over K in the following figure. We can see that the Pass@K difference between DS-GRPO with base model does not go down as K increases. This shows that DS-GRPO truly mitigates diversity collapse. The figure is shown in Fig 10.

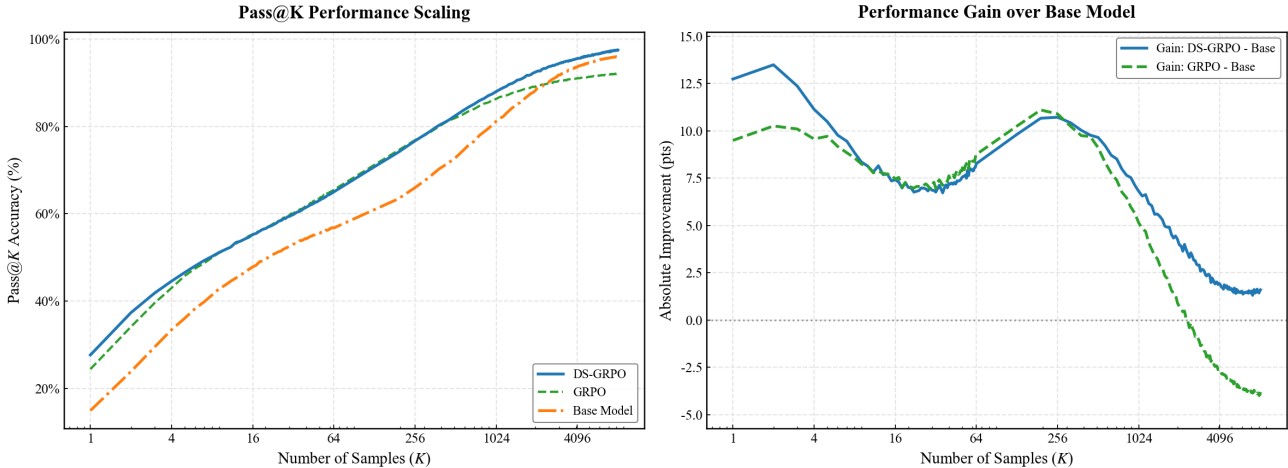

*Figure 10.* Pass@K change of GRPO and DS-GRPO.

### F.2. Comparison of Differential Entropy Control with Other Entropy Based Methods

We extended our experiments to four mathematical reasoning datasets, comparing our method against Entropy Bonus, Entropy Penalty, and vanilla GRPO. The results are shown in Figure 11, our proposed method outperforms competing baselines in the vast majority of settings. The results highlight the performance gain of our Differential Entropy method over vanilla GRPO and other entropy-based variants. Notably, our method exhibits consistent performance gains across varying $K$, particularly demonstrating superior scaling capability at higher $K$ values (e.g., $K \geq 4$).

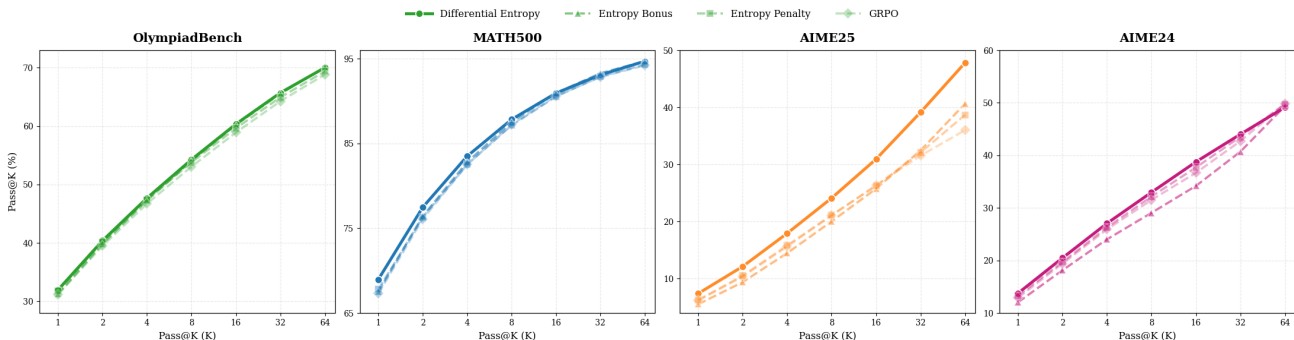

*Figure 11.* Differential entropy control vs entropy bonus, entropy penalty, and vanilla GRPO on four math benchmarks (OlympiadBench, MATH500, AIME25, AIME24; Qwen2.5-Math-1.5B).

### F.3. Statistical Analysis of DS-GRPO Improvements

We further quantify the performance gains of DS-GRPO compared to the GRPO baseline. The improvements, averaged across the four base models, are detailed in Table 2. The largest gains concentrate on the more challenging AIME benchmarks—particularly AIME 2025, where DS-GRPO improves Pass@64 by $9.4\%$ on average. On the easier benchmarks (MATH500, OlympiadBench) the gains are smaller but consistently positive across all $K$. On AMC 2023, the gain vanishes at large $K$ as both methods saturate near $100\%$.

*Table 2.* Average performance improvement of DS-GRPO over GRPO across different datasets. The values represent the percentage point increase in Pass@K.

| Dataset | P@1 | P@2 | P@4 | P@8 | P@16 | P@32 | P@64 |
|---|---|---|---|---|---|---|---|
| MATH500 | +0.9% | +0.9% | +0.8% | +0.7% | +0.7% | +0.8% | +0.9% |
| AIME 2024 | -0.4% | -0.4% | +0.1% | +0.9% | +1.8% | +2.9% | +3.9% |
| AIME 2025 | +0.3% | +0.6% | +1.2% | +2.2% | +4.0% | +6.4% | +9.4% |
| OlympiadBench | +0.5% | +0.4% | +0.4% | +0.5% | +0.6% | +0.8% | +1.0% |
| AMC 2023 | +0.6% | +0.6% | +0.5% | +0.2% | -0.1% | -0.3% | -0.5% |

