# OpenReview forum: "Differential Smoothing Mitigates Sharpening and Improves LLM Reasoning"
_ICML.cc/2026/Conference — ICML 2026 regular_

### Official Review · Reviewer_NNxA · 2026-03-08

**Soundness:** 3
**Presentation:** 3
**Significance:** 4
**Originality:** 4
**Overall Recommendation:** 4
**Confidence:** 3

**Summary:**

The paper solves one of the big problems in GRPO called diversity collapse. Prior methods have been able to solve this problem, but hurt the pass@1 performance of the model. The paper produces a novel differential smoothing algorithm that outperforms GRPO at both pass@1 and pass@k. The paper shows improvement over GRPO both theoretically and empirically.

**Compliance With Llm Reviewing Policy:**

Affirmed.

**Key Questions For Authors:**

- What is the reason why the improvements in the smaller models are minimal (especially shown in Figure 3)?

**Strengths And Weaknesses:**

Strengths:
- The paper is well written and easy to follow.
- The paper is well motivated, as several papers show that GRPO reduces pass@k accuracy.
- The results are strong and scale well.

Weakness:
- Please write better captions, especially for Figure 1. Also Figure itself can be made better.
- The additional computational cost relative to GRPO should be reported.
- Both the problem and solution are confined to GRPO or to other RLVR methods since the evaluation is only limited to GRPO.
- Hyperparameter analysis on $\gamma_{n}$ and $\gamma_{p}$ are missing.

---

> ### Author Rebuttal · Authors · 2026-03-30
>
> ### Weakness 1: Better captions and figures.
>
> Thank you for the suggestion. We will improve the captions (especially for Figure 1) and the figure design in the revised version.
>
> ### Weakness 2: Additional computational cost.
>
> **DS-GRPO introduces *no additional computational cost* over GRPO.**
>
> - **Computing $\log \pi\_{\theta\_{\text{old}}}$ requires no extra forward pass.** In standard GRPO (Section 3.1), the policy already decodes a group of $G$ completions $\{y_i\}\_{i=1}^{G} \sim \pi\_{\theta_{\text{old}}}(\cdot \mid x)$ and computes $\log \pi\_{\theta\_{\text{old}}}(y\_i \mid x)$ for the importance ratio $\rho_{i,t}$. DS-GRPO reuses these same log-probabilities to construct the modified advantage $A\_i^{\text{DS}}$ (Eq. 3.2), so *no additional forward pass or sampling* is needed.
>
> - **DS-GRPO does not need extra training time.** We report the training time per steps of DS-GRPO and GRPO across all our experimental settings below. The overhead is consistently within 3%, well within normal run-to-run variance.
>
> | Task | Model | GRPO | DS-GRPO | Overhead |
> |------|-------|------|---------|----------|
> | Countdown | Qwen2.5-3B | 55.4 | 54.2 | -2.2% |
> | Math | Qwen3-1.7B | 204.1 | 209.8 | +2.8% |
> | Math | Ministral-8B-Instruct | 117.3 | 117.4 | +0.1% |
> | Math | Qwen2.5-Math-1.5B | 52.5 | 52.6 | +0.2% |
> | Math | Qwen2.5-Math-7B | 83.4 | 86.9 | +4.2% |
>
> *Table: Wall-clock training time (seconds) of GRPO and DS-GRPO across all experimental settings.*
>
> ### Weakness 3: Limitation to RLVR
>
> **Our method also generalizes to RLHF.** We train Llama-3.2-3B-Instruct on 179k RLHF prompts using an inherently noisy 8B reward model. As shown below, DS-GRPO still achieves a 15.8% relative improvement over GRPO on Arena-Creative-Writing, confirming effectiveness under noisy rewards.
>
> | Method | Arena-Creative-Writing |
> |--------|----------------------|
> | Llama-3.2-3B-Instruct (base) | 5.2% |
> | GRPO | 20.2% |
> | **DS-GRPO** | **23.4%** (+15.8%) |
>
> *Table: RLHF results (win rate) on Arena-Creative-Writing.*
>
> ### Weakness 4: Analysis on Hyperparameter
>
> **We have analysis on hyperparameter in Section F.1 of our paper.**
>
> ### Question 1: Improvement on smaller models
>
> We believe the smaller improvement on smaller models can be attributed to two factors:
>
> - **Limited capacity to maintain diverse reasoning paths.** Generating diverse solutions requires the model to sustain multiple modes in its output distribution. Smaller models lack the representational capacity to support this, so even with the entropy bonus encouraging diversity, they cannot effectively learn and represent genuinely different reasoning strategies.
>
> - **Smaller models are still learning correctness; diversity is secondary.** For models that are still struggling to improve basic accuracy, most of the learning capacity is devoted to mastering correct reasoning. Larger models have already achieved strong correctness and thus have more "room" to benefit from diversity improvements.
>
> That said, DS-GRPO *consistently* outperforms GRPO on smaller models across all values of $K$ without any degradation, demonstrating that our method is safe and robust across different model scales.

---

> > ### Author Rebuttal · Reviewer_NNxA · 2026-04-01
> >
> > Thanks for the response. Most of the concerns are resolved. Except for when I said that method was not shown in any other RLVR method besides GRPO, I meant methods that use group inference to calculate relative advantages.
> > I am not sure whether the same problem (the one the paper tried to solve) holds for RLHF.
> > Further, I would maintain my positive score.

---

> > > ### Author Response · Authors · 2026-04-04
> > >
> > > Thank you for the follow-up. We agree that it is important to verify whether differential smoothing applies to other RLVR methods beyond GRPO. To address this, we apply our method to RLOO, another representative method that uses group inference to compute relative advantages.
> > >
> > > We train Qwen2.5-3B on Countdown using both RLOO and DS-RLOO (our differential smoothing adaptation). As shown below, DS-RLOO consistently improves over RLOO across all values of $K$ (+5.0pp on Pass@1, +2.9pp on Pass@64), confirming that (1) diversity collapse also occurs in RLOO, and (2) our differential smoothing principle generalizes beyond GRPO to other group-based RLVR methods.
> > >
> > > | Method | P@1 | P@2 | P@4 | P@8 | P@16 | P@32 | P@64 |
> > > |--------|-----|-----|-----|-----|------|------|------|
> > > | RLOO | 51.0 | 57.0 | 60.6 | 63.5 | 65.9 | 68.0 | 70.0 |
> > > | **DS-RLOO** | **56.0** | **62.1** | **65.5** | **67.8** | **69.9** | **71.4** | **72.9** |
> > >
> > > *Table: Countdown results (Pass@K) comparing RLOO and DS-RLOO on Qwen2.5-3B.*

---

### Official Review · Reviewer_Vxuc · 2026-03-08

**Soundness:** 3
**Presentation:** 3
**Significance:** 3
**Originality:** 3
**Overall Recommendation:** 4
**Confidence:** 4

**Summary:**

This paper proposes a novel method to mitigate diversity collapse during RL post-training. Specifically, it first identifies the root causes of diversity collapse in existing RL fine-tuning approaches, namely selection bias and reinforcement bias. Based on this analysis, the proposed method applies an entropy bonus exclusively to positive samples and an entropy penalty exclusively to negative samples. Moreover, the paper provides theoretical insights into why the proposed approach can improve both diversity (i.e., variance) and accuracy (i.e., pass @1). Extensive experiments further validate the effectiveness of the method.

**Compliance With Llm Reviewing Policy:**

Affirmed.

**Ethical Review Concerns:**

Not applied

**Final Justification:**

Thanks for the author's rebuttal, I will keep my score.

**Key Questions For Authors:**

1) The proof of Prosposition 6 at page 15 seems a little unclear, why we consider the Pass N rather than Pass 1 when we try to proof the selection bias? Does this proposition still holds if we consider Pass 1 only?


2) For the proof of Theorem~4, which part of the appendix demonstrates that the variance (i.e., diversity) is significantly larger? It seems that the proof mainly shows that, by properly choosing the hyperparameter $\gamma_{ds}$, the policy $\pi_{ds}$ can achieve the same accuracy as $\pi_{ent}$ while maintaining a smaller or equal KL divergence. I would suggest that the authors clarify this proof and make the logical connection to the diversity claim more explicit.

**Limitations:**

yes

**Strengths And Weaknesses:**

Strengths:

1) The proposed method is well-motivated by theory, and it achieves superior results compared to either the original GRPO baseline or other entropy-based methods.

2) The theory is elegant, especially in analyzing why existing RLHF suffers from the entropy collapse issue and how the proposed method can improve diversity and performance at the same time.

3) The authors evaluate the effectiveness of their proposed method on several math benchmarks (e.g., AIME 2024 and 2025), achieving improvements in both pass@1 and pass@k compared with the selected baselines.

Weakness:

1) Theory: please address my questions regarding the rigiorous and correctness of the proof. Otherwise, it looks good to me.

2) Experiments: I feel that this method is fairly general and not limited to the math domain. It would be great if the authors could demonstrate the generalizability of the proposed approach in other domains as well (e.g., code).

---

> ### Author Rebuttal · Authors · 2026-03-30
>
> Thank you for your detailed review. We address your questions below.
>
> ### Question 1/Weakness 1: Proof of Proposition 6
>
> **Our proof actually concerns Pass@1, not Pass@N.** In the proof, $N$ denotes the number of times a trajectory is independently sampled during training (i.e., the group size $G$ in GRPO). We use $N$ to characterize whether a correct trajectory $\tau$ is *discovered* during training: it receives a positive reward if it appears in at least one of $N$ samples, which occurs with probability $1 - (1-\pi_{\text{base}}(\tau))^N$. Since this probability is monotonically increasing in $\pi_{\text{base}}(\tau)$, trajectories with higher base probability are more likely to be discovered and subsequently reinforced. This is the selection bias. Crucially, the consequence of this bias is that $\pi_{\text{van}}(\tau)$---the probability of sampling $\tau$ in a *single* generation---increases more for high-probability trajectories. Therefore, our analysis directly characterizes the bias in Pass@1.
>
> ### Question 2/Weakness 1: Theoretical Analysis on Diversity
>
> Thank you for your question. We will clarify this in the revised paper. The proof of Theorem 4 proceeds in two steps: In Lemmas 10–12, we first show that $\pi_{\text{DS}}$ can achieve the *same* correctness and variance (diversity) as $\pi_{\text{ent}}$ with *smaller or equal* KL divergence (the variance/diversity claim is proved at the end of each lemma, e.g., lines 957–970 for Lemma 10). Then, since $\pi_{\text{DS}}$ has remaining KL budget, we can further adjust parameters to achieve *higher* correctness and diversity under the same KL constraint.
>
> ### Weakness 2: Generalizability to other domains (e.g., code).
>
> **Our method generalizes to code tasks.** We train Qwen3-0.6B on CruxEval for 400 steps and evaluate on a held-out test split (80 examples). As shown below, DS-GRPO improves over GRPO across all $K$ (+1.3pp on Pass@1, +1.0pp on Pass@64), confirming that our method generalizes beyond mathematical reasoning.
>
> | Method | P@1 | P@2 | P@4 | P@8 | P@16 | P@32 | P@64 |
> |--------|-----|-----|-----|-----|------|------|------|
> | GRPO | 34.5 | 38.5 | 42.3 | 45.9 | 49.3 | 52.5 | 55.4 |
> | **DS-GRPO** | **35.8** | **39.8** | **43.6** | **47.2** | **50.5** | **53.7** | **56.4** |
>
> *Table: Code understanding results (Pass@K) on CruxEval-Test.*
>
> ---

---

> > ### Author Rebuttal · Reviewer_Vxuc · 2026-04-01
> >
> > Thank you for your rebuttal. I still have some concerns regarding the theoretical analysis of diversity. Could you please indicate which line or equation corresponds to the variance? I was able to identify the parts related to accuracy and KL divergence in the proof.

---

> > > ### Author Response · Authors · 2026-04-03
> > >
> > > Thank you for the follow-up. We point to the exact lines and explain the variance analysis below.
> > >
> > > - **Proof structure.** The proof proceeds in two steps: (1) In Lemmas 10–13, we show that $\pi_{\text{DS}}$ achieves the *same* correctness and variance as $\pi_{\text{ent}}$ with *smaller* KL divergence. (2) Since $\pi_{\text{DS}}$ has remaining KL budget, we can adjust both $\tilde{\gamma}$ and $\beta_{\text{DS}}$ to improve both correctness and variance no worse than vanilla RL under the same KL constraint.
> > >
> > > - **Location of variance proof.** The variance analysis appears at the end of each lemma: Lemma 10 (lines 957–970), Lemma 11 (lines 1040–1051), Lemma 12 (lines 1117–1128), and Lemma 13 (lines 1187–1197).
> > >
> > > - **Detailed walkthrough (Lemma 10, lines 957–970).**
> > >   1. Lines 959–961: By setting $\tilde{\gamma} = \gamma_{\text{DS}}/\beta_{\text{DS}} = \gamma_{\text{ent}}/\beta_{\text{ent}}$, the policy $\pi_{\text{DS}}$ assigns each correct trajectory $\tau \in \mathcal{C}$ probability proportional to $[\pi_{\text{base}}(\tau)]^{1-\tilde{\gamma}} \exp(r(\tau)/\beta_{\text{DS}})$.
> > >   2. Lines 965–967: Similarly, $\pi_{\text{ent}}^*$ assigns each correct trajectory probability proportional to $[\pi_{\text{base}}(\tau)]^{1-\tilde{\gamma}} \exp(r(\tau)/\beta_{\text{ent}})$.
> > >   3. Since all correct trajectories share $r(\tau) = 1$, the $\exp(\cdot)$ term is a constant within each policy. Both policies therefore assign probability *proportional to* $[\pi_{\text{base}}(\tau)]^{1-\tilde{\gamma}}$ over correct trajectories, i.e., the relative distribution is identical.
> > >   4. Since $\sigma$ (Definition 3) depends only on this relative distribution, $\sigma(\pi_{\text{DS}}) = \sigma(\pi_{\text{ent}})$ follows immediately (line 970).
> > >
> > >   The same argument applies to Lemmas 11–13.
> > >
> > > ---

---

### Official Review · Reviewer_NJaH · 2026-03-11

**Soundness:** 3
**Presentation:** 3
**Significance:** 3
**Originality:** 3
**Overall Recommendation:** 4
**Confidence:** 4

**Summary:**

To addresses "diversity collapse" in RL fine-tuned LLMs, authors theoretically prove this collapse is driven by selection and reinforcement biases that disproportionately amplify already high-probability correct answers. To resolve the inherent trade-off between correctness and diversity, the authors propose Differential Smoothing (DS), which applies an entropy bonus exclusively to correct trajectories to promote solution variety, and an entropy penalty to incorrect ones to drive accuracy. Based on the evaluations across various models and reasoning benchmarks, this approach consistently improves both Pass@1 and Pass@K simultaneously.

**Compliance With Llm Reviewing Policy:**

Affirmed.

**Final Justification:**

Thanks for authors' detailed rebuttal, which have addressed all of my previous concerns (especially the additional experiments on multimodal RL fine-tuning). Overall, I believe this is a well-motivated paper and recommend it for acceptance.

**Key Questions For Authors:**

See `Weakness` part.

**Limitations:**

Yes.

**Strengths And Weaknesses:**

## Strengths
1. The paper is generally well-written and easy to follow, with a clear description of the method.
2. This paper conduct comprehensive experiments to validate the effectiveness of proposed method.
3. This paper provides intuitive demonstration cases on the effect of entropy to help understand how the proposed method affects RL finetuining.

## Weaknesses
1. **Evaluation of "Reasoning Diversity"**

The paper relies heavily on `Pass@1` and `Pass@K` metrics to claim improvements in reasoning and diversity. However, `Pass@K` primarily measures the success rate and breadth of final outcomes, which does not necessarily equate to the diversity of the intermediate reasoning trajectories. The paper's significance would be greatly enhanced by including a semantic diversity evaluation of the intermediate generation steps (e.g., clustering the reasoning paths of correct solutions) to prove that the model explores genuinely different problem-solving approaches.

2. **About Noisy Reward Models**

The theoretical proofs assume perfect reward estimation, and experiments rely solely on deterministic math/logic tasks with exact rule-based verifiers. In real-world RL fine-tuning (e.g., preference alignment), Reward Models (RMs) are inherently noisy. Applying an entropy bonus to a false-positive trajectory or a penalty to a false-negative one could severely degrade performance. Could the authors further clarify how Differential Smoothing handles noisy rewards?

3. **Generalization to Multimodal RL Finetuning**

While the paper claims broad applicability for mitigating diversity collapse, the empirical evaluation is strictly limited to text-based reasoning tasks. It remains unclear whether the proposed differential objective remains effective in more complex, high-dimensional spaces like Large Multimodal Model (LMM) reasoning.

I'll be more than willing to adjust my evaluation based on authors' responses.

---

> ### Author Rebuttal · Authors · 2026-03-30
>
> Thank you for your question. We now address your concern as follows:
>
> ### Weakness 1: Evaluation of "Reasoning Diversity"
>
> Thank you for this suggestion. We agree that evaluating diversity beyond Pass@K is valuable, and we provide additional evidence below.
>
> **On the Countdown task, distinct final answers directly reflect distinct reasoning paths.** Unlike tasks where multiple reasoning paths may converge to the same answer, in Countdown each correct solution corresponds to a unique arithmetic expression (e.g., $(75+25)\times 4 - 10$ vs. $75\times 4 + 25 - 10$). Therefore, counting the number of *distinct correct solutions* is a faithful measure of reasoning trajectory diversity for this task.
>
> We sampled 32 completions per question on the Countdown test set and computed the average number of distinct correct solutions found. As shown below, DS-GRPO discovers 14.1% more distinct solutions than GRPO, confirming that DS-GRPO explores genuinely different problem-solving strategies rather than merely repeating the same high-probability path.
>
> | Method | Avg. Distinct Correct Solutions |
> |--------|-------------------------------|
> | GRPO | 1.42 |
> | **DS-GRPO** | **1.62** (+14.1%) |
>
> *Table: Average number of distinct correct solutions per question (32 samples) on the Countdown test set.*
>
> ### Weakness 2: How does Differential Smoothing handle noisy rewards?
>
> Thank you for this important question. We address it from both a theoretical and an empirical perspective.
>
> - **Differential Smoothing is inherently more robust to false positives than vanilla RL.** The entropy bonus on correct samples flattens the probability distribution over all positively-rewarded trajectories, preventing the model from concentrating too much probability mass on any single one. When a noisy reward model produces a false positive (an incorrect trajectory mislabeled as correct), vanilla RL may sharply reinforce this trajectory, assigning it disproportionately high probability. In contrast, because DS-GRPO spreads probability more evenly across all positive samples, the weight assigned to any individual false positive remains limited---making DS-GRPO more robust to reward noise.
>
> - **The entropy correction term is noise-free, diluting the impact of reward noise.** The advantage modification in DS-GRPO, $-\gamma_p \log \pi_{\theta_{\text{old}}}$, is computed entirely from the old policy's own log-probabilities and does *not* depend on the reward model at all, and will not be affected by the reward model.
>
> - **RLHF experiment with a noisy reward model confirms robustness.** We train Llama-3.2-3B-Instruct on 179k RLHF prompts using an inherently noisy 8B reward model. As shown below, DS-GRPO still achieves a 15.8% relative improvement over GRPO on Arena-Creative-Writing, confirming effectiveness under noisy rewards.
>
> | Method | Arena-Creative-Writing |
> |--------|----------------------|
> | Llama-3.2-3B-Instruct (base) | 5.2% |
> | GRPO | 20.2% |
> | **DS-GRPO** | **23.4%** (+15.8%) |
>
> *Table: RLHF results (win rate) on Arena-Creative-Writing.*
>
> ### Weakness 3: Generalization to multimodal RL fine-tuning.
>
> **Our method can also generalize to multimodal RL fine-tuning.** Thank you for this suggestion. We conduct a new experiment on multimodal reasoning to directly address this concern. We train Qwen2.5-VL-3B-Instruct on Geo3K (geometry problems with diagram inputs) for 700 steps. As shown below, DS-GRPO consistently improves over GRPO across all values of $K$ (+2.3pp on Pass@1, +1.8pp on Pass@64), confirming that differential smoothing generalizes to multimodal settings where the policy must jointly reason over visual and textual inputs.
>
> | Method | P@1 | P@2 | P@4 | P@8 | P@16 | P@32 | P@64 |
> |--------|-----|-----|-----|-----|------|------|------|
> | GRPO | 45.9 | 52.8 | 58.8 | 63.6 | 67.6 | 70.6 | 72.2 |
> | **DS-GRPO** | **48.2** | **55.2** | **61.2** | **66.0** | **69.8** | **72.4** | **74.0** |
>
> *Table: Multimodal reasoning results (Pass@K) on Geo3K-Test.*
>
> ---

---

> > ### Author Rebuttal · Reviewer_NJaH · 2026-04-01
> >
> > Thanks for authors' detailed rebuttal, which have addressed all of my previous concerns (especially the additional experiments on multimodal RL fine-tuning). Overall, I believe this is a well-motivated paper and recommend it for acceptance.

---

> > > ### Author Response · Authors · 2026-04-06
> > >
> > > We sincerely appreciate you reconsidering your evaluation, and we thank you for the positive score on our work.

---

### Official Review · Reviewer_yfBd · 2026-03-12

**Soundness:** 3
**Presentation:** 2
**Significance:** 2
**Originality:** 3
**Overall Recommendation:** 4
**Confidence:** 4

**Summary:**

This paper investigates the "diversity collapse" problem in RL fine-tuning for LLMs. The authors find that vanilla RL makes the model too "sharp" on high-probability correct answers. To solve this, they propose a new method called Differential Smoothing (DS). The core idea is very interesting: using entropy bonus on correct samples to encourage diversity, while using entropy penalty on incorrect samples to improve correctness. Theoretical proofs and experiments on Countdown and MATH datasets are provided.

**Compliance With Llm Reviewing Policy:**

Affirmed.

**Final Justification:**

Thanks for your reply and I have raised my rating. Good work!

**Key Questions For Authors:**

See above.

**Limitations:**

Yes.

**Strengths And Weaknesses:**

The logic is very clear. The authors first give a formal proof of why diversity collapse happens (Selection and Reinforcement Bias), which provides a solid foundation for the proposed method.
The method is simple but effective. Only modifying the reward function can achieve both Pass@1 and Pass@K improvement. It is easy to implement in existing GRPO frameworks.
Extensive experiments. The paper tests many models from 1B to 7B, and the performance gain on AIME24 (up to 6.7%) is quite significant. It proves the robustness of DS-GRPO.

However,
1. The paper uses $\gamma_p$ and $\gamma_n$ to control the smoothing. However, how to choose these parameters for a new task is not very clear. Is there a general rule or we must do grid search every time?

2. Although the paper says it is efficient, the calculation of $\log \pi_{\theta_{old}}$ for every trajectory might increase the training time. The authors should provide more data about the training cost comparison.

---

> ### Author Rebuttal · Authors · 2026-03-30
>
> Thank you for your question. We now address your concern as follows:
>
> ### Weakness 1: How to choose $\gamma_p$ and $\gamma_n$?
>
> For a new task, we can consider the following factors to choose $\gamma_p$ and $\gamma_n$:
>
> - **Select based on task's sensitivity to entropy and Solution Multiplicity.** As discussed in Section 3.7.2, the effectiveness of increasing vs. decreasing entropy is task-dependent and can be characterized by *Solution Multiplicity* (Eq. 2). Specifically, for tasks with high Solution Multiplicity (e.g., Countdown), diversity gains outweigh the correctness trade-off, so a larger $\gamma_p$ is preferred; for tasks with low Solution Multiplicity (e.g., MATH500), correctness gains are more valuable, so a larger $\gamma_n$ is preferred. Solution Multiplicity can be estimated from a small held-out set before training, providing a practical and principled starting point without any grid search.
>
> - **Our method is not sensitive to $\gamma_n$ and $\gamma_p$, so we can do grid search on a small range of values.** Our parameter sensitivity analysis (Appendix F.1, Figure 10 and Table 2) shows that DS-GRPO is robust across a wide range of $(\gamma_p, \gamma_n)$ combinations: Pass@1 varies between 0.57–0.67 and Pass@64 between 0.78–0.84 over the entire grid. This means that even a rough estimate from the above principle yields near-optimal performance, and a lightweight grid search over a small range suffices to fine-tune the values if needed.
>
> ### Weakness 2: Training cost comparison.
>
> Thank you for raising this point. We clarify that DS-GRPO introduces *no meaningful computational overhead* over vanilla GRPO:
>
> - **Computing $\log \pi_{\theta_{\text{old}}}$ requires no extra forward pass.** In standard GRPO (Section 3.1), the policy already decodes a group of $G$ completions $\{y_i\}\_{i=1}^{G} \sim \pi\_{\theta\_{\text{old}}}(\cdot \mid x)$ and computes $\log \pi_{\theta\_{\text{old}}}(y\_i \mid x)$ for the importance ratio $\rho_{i,t}$. DS-GRPO reuses these same log-probabilities to construct the modified advantage $A\_i^{\text{DS}}$ (Eq. 3.2), so *no additional forward pass or sampling* is needed.
>
> - **DS-GRPO does not need extra training time.** We report the training time per steps of DS-GRPO and GRPO across all our experimental settings in the table below. The overhead is consistently within 3%, well within normal run-to-run variance.
>
> | Task | Model | GRPO | DS-GRPO | Overhead |
> |------|-------|------|---------|----------|
> | Countdown | Qwen2.5-3B | 55.4 | 54.2 | -2.2% |
> | Math | Qwen3-1.7B | 204.1 | 209.8 | +2.8% |
> | Math | Ministral-8B-Instruct | 117.3 | 117.4 | +0.1% |
> | Math | Qwen2.5-Math-1.5B | 52.5 | 52.6 | +0.2% |
> | Math | Qwen2.5-Math-7B | 83.4 | 86.9 | +4.2% |
>
> *Table: Wall-clock training time (seconds) of GRPO and DS-GRPO across all experimental settings.*
>
> ---

---

> > ### Author Rebuttal · Reviewer_yfBd · 2026-04-02
> >
> > Thanks for your reply and I have raised my rating. Good work!

---

> > > ### Author Response · Authors · 2026-04-06
> > >
> > > Thank you very much on the positive rating of our work. We will also update in the revised version of our paper.

---

### Decision · Program_Chairs · 2026-04-30

**Decision:**

Accept (regular)

**Comment:**

All reviewers agree that the paper attacks an interesting problem (diversity collapse in RLVR) with a novel solution (applying entropic bonuses to correct answers and penalties to incorrect answers).  While one of the reviewers was initially concerned with the rigor of the theory, it seems that these concerns have been addressed.  Moreover, the reviewers all agree that sufficient empirical examples have been conducted, with one reviewer asking for a multi-modal example, which was then provided.